# FailureSensorIQ: A Multi-Choice QA Dataset for Understanding Sensor Relationships and Failure Modes

**Christodoulos Constantinides**[2*]    **Dhaval Patel**[1*]    **Shuxin Lin**[1*]    **Claudio Guerrero**[2]
**Sunil Dagajirao Patil**[2]    **Jayant Kalagnanam**[1]
[1]IBM TJ Watson Research Center    [2]IBM
{christodoulos.constantinides, pateldha@us,shuxin.lin,claudio.guerrero,
patil.sunil@in,jayant@us}@ibm.com
*Equal contribution.

## Abstract

We introduce FailureSensorIQ, a novel Multi-Choice Question-Answering (MCQA) benchmarking system designed to assess the ability of Large Language Models (LLMs) to reason and understand complex, domain-specific scenarios in Industry 4.0. Unlike traditional QA benchmarks, our system focuses on multiple aspects of reasoning through failure modes, sensor data, and the relationships between them across various industrial assets. Through this work, we envision a paradigm shift where modeling decisions are not only data-driven using statistical tools like correlation analysis and significance tests, but also domain-driven by specialized LLMs which can reason about the key contributors and useful patterns that can be captured with feature engineering. We evaluate the Industrial knowledge of over a dozen LLMs including GPT-4, Llama, and Mistral on FailureSensorIQ from different lens using Perturbation-Uncertainty-Complexity analysis, Expert Evaluation study, Asset-Specific Knowledge Gap analysis, ReAct agent using external knowledge-bases. Even though closed-source models with strong reasoning capabilities approach expert-level performance, the comprehensive benchmark reveals a significant drop in performance that is fragile to perturbations, distractions, and inherent knowledge gaps in the models. We also provide a real-world case study of how LLMs can drive the modeling decisions on 3 different failure prediction datasets related to various assets. We release: (a) expert-curated MCQA for various industrial assets, (b) FailureSensorIQ benchmark and Hugging Face leaderboard based on MCQA built from non-textual data found in ISO documents, and (c) "LLMFeatureSelector", an LLM-based feature selection scikit-learn pipeline. The software is available at `https://github.com/IBM/FailureSensorIQ`.

## 1 Introduction

In the era of agentic workflows, LLMs must not only answer domain-specific questions accurately but also generate appropriate reasoning as part of an AI agent's decision-making process. LLMs such as GPT [24], LlaMA [33], Gemini [31], Mistral [13], and others are typically pre-trained on vast corpora like CommonCrawl, Wikipedia, and books, and then fine-tuned on domain-specific datasets, including code (e.g., GitHub), biomedical data (e.g., PubMed), and scientific literature (e.g., Semantic Scholar, Arxiv). This enables LLMs to assimilate broad knowledge across diverse fields, including general knowledge, mathematics, finance, and more. However, a key question remains: Do these models possess the specialized knowledge and reasoning abilities necessary for complex domains like medicine, chemistry, biology, and industrial applications? As evident in recent literature [49], the domain-specific and general QA datasets have played a pivotal role in advancing LLM capabilities and assessing their readiness for real-world applications, however exploration in industrial domains

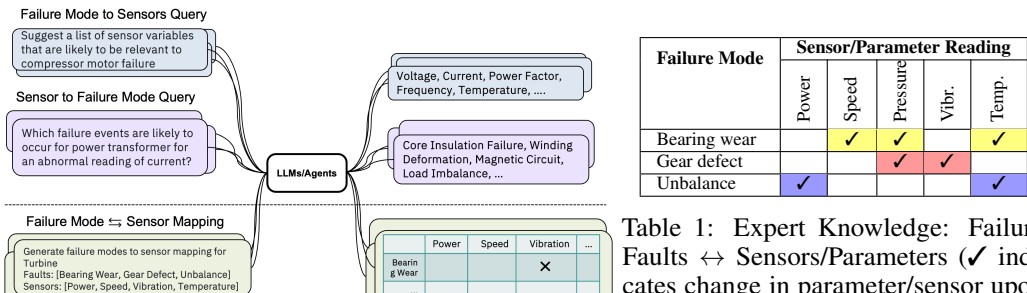

Table 1: Expert Knowledge: Failure Faults ↔ Sensors/Parameters (✓ indicates change in parameter/sensor upon failure) for turbine asset.

| Failure Mode | Power | Speed | Pressure | Vibr. | Temp. |
|---|---|---|---|---|---|
| Bearing wear | | ✓ | ✓ | | ✓ |
| Gear defect | | | ✓ | ✓ | |
| Unbalance | ✓ | | | | ✓ |

Figure 1: Example of AI Tasks for Industry 4.0 Applications

remains limited. Although some isolated efforts have addressed tasks like classifying work orders or managing maintenance issues [30], comprehensive benchmark datasets to assess LLMs on critical industrial challenges such as predictive maintenance, sensor fault detection, and asset management are still lacking.

This paper aims to address the existing gap by introducing a set of specialized benchmarks designed to assess LLMs' reasoning abilities within the context of industrial operations. In doing so, we seek to enhance their utility in domains like Industry 4.0, where domain-specific expertise and accurate, real-time decision-making are critical. A key application within Industry 4.0 is Condition-Based Maintenance (CBM), which focuses on monitoring the health of assets using sensor data. Typically, IoT devices collect data from various sensors, such as temperature, power, and pressure, which is then analyzed to predict potential failures and recommend proactive maintenance before breakdowns occur [11]. To improve failure detection, Failure Modes and Effects Analysis (FMEA), a reliability engineering methodology, is often applied to the sensor data and failure modes. This process involves establishing **relationships between an asset's potential failures and one or more monitored sensors** that may indicate these failures when anomalies in the sensor data are detected.

## 1.1 Fundamentals of Condition-Based Maintenance for Industrial Assets

Detecting maintenance needs early in the asset lifecycle is essential for proactive management and maximizing asset reliability [22, 43]. To facilitate this, effective monitoring systems need to be able to identify the specific parameters associated with each failure mode. Table 1 presents an example of a mapping table for turbines. Each row corresponds to a specific failure mode of the turbine, while each column represents a measurable parameter or sensor typically used to monitor turbine performance. Cells marked with a ✓ symbol indicate the relevant parameters that can help detect a particular failure mode. For example, the power sensor is marked as useful for detecting unbalance faults in turbines. When introducing a new asset into the CBM process, subject matter experts traditionally face the labor-intensive task of using a data-driven approach to identify the relationships between each failure mode and the corresponding sensor parameters. This process involves extensive data collection and requires a high level of expertise in asset management.

In contrast to a purely data-driven approach, could an LLM assist in automating the discovery of these mappings? Such a knowledge-driven method could leverage the internal domain expertise of LLMs to augment the knowledge of subject matter experts. As shown in Figure 1, LLMs have the potential to act as knowledge generators, offering insights into the relationships between failure modes and sensor parameters. Rather than relying exclusively on extensive data collection, an LLM-based workflow could improve decision-making by providing a contextual understanding of how failure modes are linked to the relevant sensor parameters. We consider two key diagnostic mapping tasks involving expert knowledge:

1. **Failure Modes to Sensor Relevance** (`FM2Sensor`): Identifying the most relevant sensors for detecting early signs of a given failure.

2. **Sensor to Failure Mode Relevance** (`Sensor2FM`): Understanding which failure modes can be detected early by monitoring a specific sensor.

These mappings help structure prior expert knowledge in ways that LLMs can use for reasoning in failure diagnosis tasks. The `FM2Sensor` query is useful for automating an AI agent responsible for

building anomaly detection models or assisting engineers in selecting and planning sensor installations on machinery to track early failure signs. Conversely, the `Sensor2FM` query supports root cause analysis and allows AI agents to suggest maintenance tasks proactively, reducing downtime. By automating these processes, LLMs/Agents can significantly enhance decision-making efficiency in predictive maintenance. However, a critical question remains: How well do LLMs understand the intricate relationships between assets, their failure modes, and sensor parameters?

## 1.2 Key Contributions and Insights

We introduce **FailureSensorIQ**, a multiple-choice QA benchmarking system with a dataset that explores the relationships between sensors and failure modes for 10 industrial assets. By only leveraging the information found in ISO documents [11] and expert crafted question templates, we developed a data generation pipeline that creates questions in two formats: (i) row-centric (`FM2Sensor`) and (ii) column-centric (`Sensor2FM`). Additionally, we designed questions in a **selection** vs. **elimination** format, taking advantage of the fact that the absence of an ✓ in a cell (as shown in Table 1) indicates irrelevant information. The **FailureSensorIQ** dataset consists of **8,296** questions across 10 assets, with 2,667 single-correct-answer questions and 5,629 multi-correct-answer questions.

The true challenge of the FailureSensorIQ dataset becomes evident through three key observations that underscore its exceptional difficulty. First, the top-10 performing models which consists of a range of frontier and large open-source models averages just 53.5% accuracy on 2,667 single-correct-answer multiple-choice questions, a result that is widely regarded as a hallmark of "hard" datasets [28, 37, 39]. Second, even models fine-tuned on datasets with diverse knowledge like HotpotQA [45] struggle, achieving only 29% accuracy on our dataset (See Appendix E.4). Third, our unified **Perturbation–Uncertainty–Complexity** analysis reveals significant performance degradation under three stressors: (i) perturbing the question results in a 5–20% performance drop (measured via ACC@Consist [14]); (ii) model uncertainty increases significantly, with an average of three options needing to be selected to achieve 90% coverage; and (iii) increasing the number of distractor options caps the maximum achievable accuracy at 12%.

The complexity of the dataset underscores the critical role that **reasoning strategies** play in shaping model's performance. This is evident from the fact that, although the top-3 models vary across different performance metrics on our Hugging Face leaderboard, all consistently exhibit implicit reasoning capabilities. To investigate this further, we evaluated different prompting methods on a set of 2,667 single-answer multiple-choice questions using a range of large open-source models. These strategies, namely Chain-of-Thought [38], Role-Playing Chain-of-Thought [19], and Self-Plan [36] revealed a compelling trend: while larger models tend to perform better overall, medium-sized models (around 70 billion parameters) experience significant performance gains via effective prompting.

Beyond reasoning complexity, **expert evaluation** on a curated subset of the dataset and analysis of asset-centric external knowledge offer deeper insights into model reliability and underlying **knowledge gaps**. Human experts achieved a maximum accuracy of 66.19% and a mean accuracy of 60.20% across three participants, underscoring the intrinsic difficulty of the task even for domain specialists. Given the vast landscape of industrial assets and their thousands of failure modes, our asset-level performance analysis reveals a **modest correlation between the volume of external knowledge** (e.g., Wikipedia, arXiv) and an LLM's ability to answer related questions. However, the current ReAct-based Agent [46] fails to deliver the expected performance improvements, indicating a need for future research into effective reasoning and retrieval strategies.

Furthermore, we evaluate models on multiple-correct MCQAs (`MC-MCQA`), where each question has exactly two correct options. The primary challenge in this setting is not only selecting all valid answers but also avoiding reasonable distractors. We evaluate several recent models using same prompt setting as single-correct MCQAs (`SC-MCQA`), where models must directly select the correct set without being told how many answers to choose. Despite progress in language modeling, exact match scores remain low (below 21%), highlighting the difficulty of precise multi-option reasoning. This stark difference underscores the importance of knowing the number of correct answers, a piece of information that is typically unavailable in standard prompting scenarios.

Finally, we present "LLMFeatureSelect", an sklearn and LLM-based **feature selection pipeline** tool. This tool can reason around variables relationships involving Assets, Failure Modes, and Sensors. We use it on 3 real-world datasets, quantify the quality of the feature recommendations, and find promising model capabilities with room for improvement (Section 6).

# 2 Methodology

Our dataset includes two question formats aligned with TruthfulQA [18]: *Single Correct* and *Multi Correct* multiple-choice QA. `SC-MCQA` questions have one correct answer, while `MC-MCQA` questions require selecting two correct answers from five options. These questions are generated using our automated pipeline, as outlined in Figure 2.

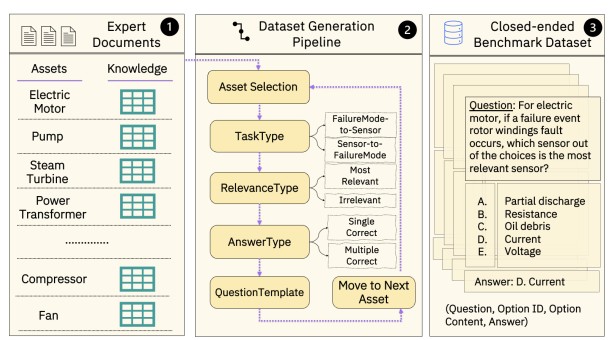
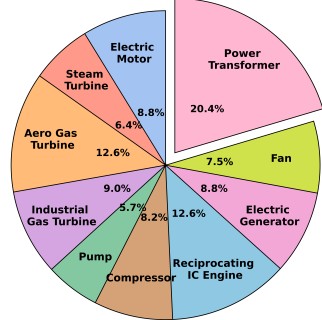

Figure 2: Multi-Choice Question Generation Pipeline. | Figure 3: Question distribution by asset.

## 2.1 Dataset Structure

Each question in the dataset is formally defined as a tuple $(Q, C, O, A)$, where:

1. $Q$: Represents the question.

2. $C$: Refers to the option ID (e.g., choices A, B, C, D and E. The number of choices are between 2 to 5).

3. $O = (o_1, o_2, \ldots, o_k)$: Represents the content of the options, assuming $Q$ has $k$ candidate options.

4. $A$: Denotes the ground-truth answer, consisting of both its ID and content.

An example prompt can be found in Appendix 10.

## 2.2 Question Generation Pipeline

The benchmark questions in our dataset are accompanied by meta-information, including asset names, task types, relevance types, and answer types. These attributes are defined during the question generation process. Note that no LLM is used to generate the dataset. Appendix B provides the pseudo-code. The process goes as follows:

1. **Asset Selection:** The pipeline begins with selecting relevant assets, such as electric motors, pumps, compressors, or other industrial equipment. These assets are defined based on standardized knowledge sources, like ISO documents.

2. **Task Type Identification:** The next step involves identifying the task type. Examples include `FM2Sensor` (failure modes to sensors) or `Sensor2FM` (sensors to failure modes).

3. **Relevance Type Determination:** Relevance type is defined based on the context of the task. This could involve selecting the *most relevant* or *irrelevant* options for the given scenario.

4. **Answer Type Definition:** Finally, the answer type is determined, specifying whether the question has a single correct answer (`SC-MCQA`) or multiple correct answers (`MC-MCQA`).

Using predefined, domain-expert-curated question templates and the above selections, closed-ended questions are generated for each asset. We have prepared approximately 10 templates per relevance and task type (see Appendix B.1). Currently, we use ISO documents [11], along with expert-curated information, to establish an initial mapping. The result is a comprehensive benchmark dataset containing well-structured questions, multiple answer options, and their corresponding correct answers. An example is shown in the last column of Figure 2. Our `SC-MCQA` corpus consists of 2,667 generated questions, although not all feature five answer options. To better understand the structure

of the corpus, we analyzed: (a) the distribution of questions by the number of answer options; (b) the frequency of selected answers across those options; and (c) the total number of questions per asset (see Figure 3). The distribution of selected options is as follows: A (752), B (729), C (491), D (408), and E (208). This imbalance in the distribution underscores the need for a robust model qualification test, which we formalize in Section 3. Such a test is essential for evaluating an LLM's tendency to favor specific answer choices and identifying emerging response patterns. The distribution of questions by the number of answer options is as follows: 2 options (487), 3 options (266), 4 options (389), and 5 options (1525). Over 50% of the questions have five answer choices, indirectly reflecting the dataset's complexity.

## 3 Perturbation-Uncertainty-Complexity Analysis

We evaluate the accuracy of several frontier and open source foundation models on the `SC-MCQA` benchmark using three settings: *Perturbation*, *Uncertainty* and *Complexity*. All these settings assume a closed-book scenario, meaning that models have no access to external information beyond the question, the prompt, and their own parameters. To ensure a diverse and representative evaluation, we select **over two dozen models** [7]. These include both closed-source models o1, o3-mini, gpt-4.1 [24], and open-source models (deepseek-r1 [9], qwen [3], granite [8], gemma [32], phi [1], mistral[13], and llama [33]).

### 3.1 Perturbed/Complex Dataset Preparation

Recent studies question whether LLMs reason before answering or justify preselected choices, revealing option biases that vary across models [4]. To address these challenges, we evaluate model performance on both the original (`SC-MCQA`) and perturbed datasets, which underwent rigorous modifications. We extend the PertEval toolkit [14] and develop two versions of the perturbed dataset: (i) `SimplePert`, which modifies the formatting of the questions by reordering the options, adding a right parenthesis to each option, and changing the option labels from A, B, C, etc., to P, Q, R, and so on. (ii) `ComplexPert`, apply all the question permutation as well as use LLM (llama-3.3-70b-instruct in this case) to change the questions also. Furthermore, to increase the question complexity, we extend each question in `SC-MCQA` to have 10 options [37]; where new choices are all distractors and then we randomized the options. We call this new dataset as `OptionsPert`. Appendix B.3 shows the pipeline that is adopted to generate the `ComplexPert` dataset.

### 3.2 Performance Study

We use a **zero-shot direct prompting** method for knowledge assessment and employ a structured output approach to simplify answer decoding after execution. It is worth noting that the prompt does not specify whether a question has a **single correct answer** or **multiple correct answers**, leaving it up to the models to infer the appropriate response based on their understanding of the task.

We evaluate model performance using several metrics: (a) **Accuracy (Acc@Original)** or **Acc**) measures the fraction of correct predictions on the original test set: $\text{Acc} = \frac{1}{|D|} \sum_{x \in D} \mathbb{I}[M(q_x) = y_x]$, where $M(q_x)$ is the model's top prediction and $y_x$ is the ground-truth label. (b) **Perturbed Accuracy (Acc@Perturb)** uses the same formula on modified queries $q_o^*(x)$. To capture robustness under ambiguity, we use (c) **Consistency-Based Accuracy (Acc@Consist)** as: $\text{Acc@Consist}(M, D) = \frac{1}{|D|} \sum_{x \in D} \mathbb{I}[M(q_x) = y_x \wedge M(q_o^*(x)) = y_o^*(x)]$, where $q_o^*(x)$ is a perturbed version of the input and $y_o^*(x)$ its corresponding label. (d) **Set Size (SS)** reflects the average number of options selected per question, indicating model selectivity. (e) **Coverage Rate (CR)** quantifies how often the correct answer appears in the selected set. (f) **Uncertainty-Adjusted Accuracy (UAcc)** combines correctness with prediction confidence, rewarding confident and accurate responses.

### 3.3 Real Knowledge Capacity Evaluation using Perturbation Analysis

Figure 4 shows performance result on `SC-MCQA` and its complex perturb version `ComplexPert` using ACC@Original, ACC@Perturb and ACC@Consist. We select top-10 performing models out of 24 entries in our leader board. From the lens of ACC@Original, "o1" is the best performing, but other frontier models are very close. We can see a significant decline in model performance when subjected to perturbed data (Wilcoxon signed-rank test with $alpha = 0.1$). For example, "o1" and "llama-4-scout" show noticeable drops in accuracy, with ACC@Original scores of 60.4% and 54.0%, falling to ACC@Perturb values of 44.2% and 24.1% respectively. Even another top-performing model, "o3-mini", only achieves an ACC@Consist of 47.5%. These results highlight the substantial

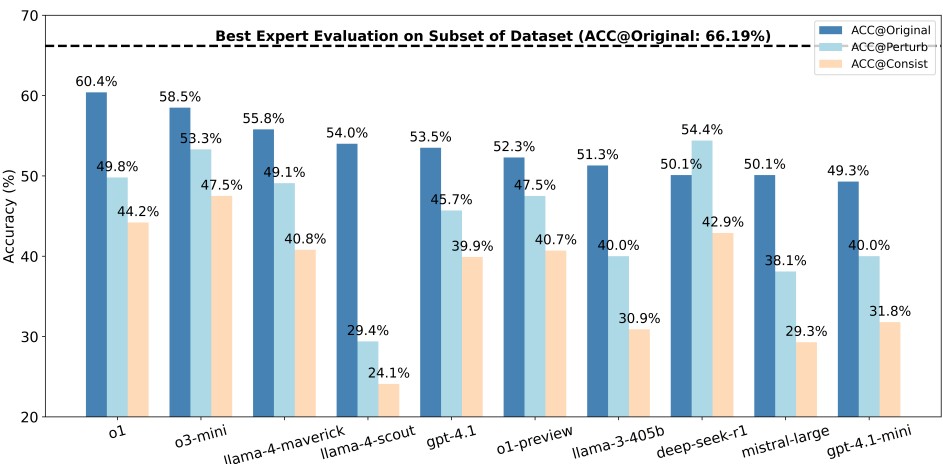

Figure 4: Real-world knowledge capacities assessed by ACC@Consist.

gap between the perceived knowledge capacity of LLMs based on static datasets and their actual ability to retain and apply knowledge under dynamic, perturbed conditions. The performance of the latest "gpt-4.1" series model is still behind reasoning driven LLMs.

Despite the drop in performance, all models demonstrate consistent mastery of certain knowledge. This is assessed by comparing each model's performance against random guessing. Given $k$ possible options (ranging from 2 to 5) with one correct answer, the expected ACC@Consist for random guessing is $\frac{1}{k^2}$. For example, with $k = 4$, the expected value would be 0.0625 (6.25%). As shown in Figure 4, the ACC@Consist values of all models are well above this random baseline. This suggests that while each model has successfully mastered some knowledge, the **models still show limited knowledge retention**, as they have ACC@Consist values below 50%, indicating they have mastered less than half of the total knowledge. This result reemphasizes the fact that an agent may probe LLMs in a different way and produce significantly different results. The performance of other models is available in Appendix E.1.

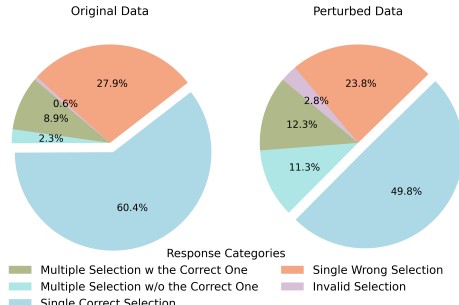

| Model | SimplePert | ComplexPert |
|---|---|---|
| o1 | -0.1204** | -0.1065** |
| o3-mini | -0.0056 | -0.0517** |
| llama-4-mav. | -0.0214** | -0.0671** |
| llama-4-scout | -0.216** | -0.2460** |
| gpt-4.1 | -0.0165 | -0.0772** |
| o1-preview | -0.0259** | -0.0480** |
| llama-3-405b | 0.0274 | -0.1121** |
| mistral-large | 0.0015 | -0.1200** |
| gpt-4.1-mini | 0.0015 | -0.1200** |

Figure 5: Response Pattern Analysis between Original Data and Complex Perturbed Data on SC-MCQA for o1.

Table 2: PDR ↑ and hypothesis test results of LLMs w.r.t. perturbation. "**": stat. significant

**Response Pattern Analysis.** Since models can select more than one options as the answer, we measure the performance of "o1" on 5 different cases. As illustrated in Figure 5, the most significant factor contributing to the performance drop is the higher frequency of selecting additional incorrect options in the perturbed data (from 11.20% to 23.6%). This behavior indicates that LLMs often choose extra incorrect options alongside the correct ones under perturbations. A potential explanation for this phenomenon could be the model's increased sensitivity to subtle variations in input, leading to over-selection.

**Performance Stability Test.** We calculate the discrepancy between the LLM's accuracy on original dataset and the perturbed dataset using Performance Drop Ratio (PDR). Table 2 shows result for

top-10 models. When PDR < 0, the perturbation decreases the overall performance of an LLM, indicating that it does not robustly acquire knowledge and skills. From a column-wise perspective, all tested LLMs show negative PDRs under the complex perturbation `ComplexPert`, indicating that these models consistently struggle to maintain performance across datasets. Furthermore, the PDR results for most models, such as "o1" and "llama-4-scout", are significantly negative, emphasizing a call for action for these models when faced with knowledge-invariant perturbations. From a row-wise perspective, the complex perturbations result in substantial performance drop for all models. These results highlight the universal vulnerability of current-generation LLMs to content-level paraphrasing and format-level changes, suggesting a pressing need for improved robustness in future models.

## 3.4 Uncertainty Quantification Analysis

To calculate uncertainty for `SC-MCQA` we follow [47]; see Appendix K for details. Our uncertainty analysis across 14 models on `SC-MCQA` (See Table 3) reveals that larger models, such as mistral-large, are better calibrated, selecting fewer answer options and achieving better performance. We get a coverage guarantee but only after a higher value of set size. In contrast, smaller models especially deepseek distill variants exhibit greater uncertainty, often over-hedging with larger set sizes and lower accuracy, despite high coverage. These trends suggest that model size correlates with confidence, and calibration remains a key challenge particularly for smaller size models, where alignment strategies may inadvertently increase ambiguity. For Industrial QA, deploying well-calibrated models (e.g., mistral-large or phi-4) is critical, while smaller models may benefit from **fine-tuning** to reduce uncertainty and improve reliability.

Table 3: Uncertainty Quantification. Model selection is constrained by **available hardware.**

| Model | UAcc ↑ | SS ↓ | CR ↑ | Acc ↑ |
|---|---|---|---|---|
| mistral-large | 50.03 | 3.02 | 91.32 | 60.83 |
| phi-4 | 46.51 | 3.02 | 91.2 | 56.39 |
| mixtral-8x22b | 41.55 | 3.36 | 93.46 | 54.75 |
| gemma-2-9b | 33.47 | 3.61 | 94.74 | 48.6 |
| granite-3.3-8b | 34.00 | 3.55 | 94.00 | 48.52 |
| granite-3.2-8b | 32.46 | 3.65 | 95.52 | 47.98 |
| mixtral-8x7b | 30.55 | 3.65 | 95.48 | 45.09 |
| qwen2.5-7b | 29.57 | 3.54 | 92.48 | 42.06 |
| granite-3.0-8b | 27.32 | 3.67 | 93.93 | 40.65 |
| llama-3.3-70b | 23.53 | 3.62 | 93.22 | 34.42 |
| dsr1-llama-70b | 20.96 | 3.81 | 95.02 | 32.48 |
| llama-3.1-8B | 19.94 | 3.83 | 94.94 | 31.15 |
| dsr1-qwen-7b | 17.76 | 3.73 | 94.04 | 26.87 |
| dsr1-llama-8b | 16.38 | 3.94 | 95.95 | 26.32 |

Table 4: Multi-choice question complexity analysis for various models including top-10.

| Model | Acc ↑ | CR ↑ | SS ↓ |
|---|---|---|---|
| llama-3-1-405b | 10.69 | 37.35 | 3.51 |
| mistral-large | 7.8 | 33.41 | 3.06 |
| llama-4-mav-17b-128e | 7.72 | 32.02 | 2.68 |
| deepseek-r1 | 6.75 | 46.98 | 3.71 |
| llama-4-scout-17b-16e | 6.34 | 79.83 | 7.77 |
| o1 | 5.74 | 41.43 | 3.72 |
| o1-preview | 5.62 | 41.84 | 3.73 |
| o3-mini | 5.62 | 41.66 | 3.72 |
| gpt-4.1 | 5.62 | 41.92 | 3.74 |
| deepseek-r1-llama-8b | 8.32 | 28.83 | 2.65 |
| granite-3.2-8b | 5.66 | 50.73 | 4.82 |
| granite-3.3-8b | 5.55 | 44.51 | 4.23 |
| gpt-4.1-nano | 5.51 | 41.39 | 3.71 |

## 3.5 Question Complexity Analysis

We use the `OptionsPert` dataset to assess question complexity. As discussed in Section 3.1, the purpose of this dataset is to reduce the likelihood of random guessing and enable systematic evaluation of model robustness under increased ambiguity. Table 4 presents the results. The top-performing model "o1" experiences a sharp accuracy drop from 60% to just 5.74% when distractor options are introduced, highlighting models' vulnerability to increased choice complexity. This mirrors real-world scenarios, such as Kaggle challenges, where selecting the relevant parameters from more than 10 options is common. These findings underscore the need for models with improved uncertainty calibration and adaptive reasoning strategies to perform reliably in complex settings.

## 3.6 Knowledge Gap Study for Industrial Assets

Each question is associated with several meta-data such as asset type (see Section 2.2). This enables us to obtain the asset-centric performance of the model. Table 5 presents the performance of the `o1` model on the `SC-MCQA` benchmark across various **asset types**, sorted by increasing ACC@Original. The results reveal substantial variation in model accuracy. Assets like Steam Turbine (43.59%) and Electric Motor (43.59%) exhibit the lowest scores, suggesting difficulties in reasoning over their operational principles. Given that large LLMs are generally trained on broad open-domain corpora, we investigated whether access to such content correlates with task performance. Specifically, we

compare model accuracy per asset type with the number of documents retrieved from trusted open-access repositories such as Arxiv, CrossRef, Wiki and Google Search. A scatter plot on a logarithmic scale reveals a mild positive correlation (See Figure 6): asset types with richer public documentation tend to yield higher accuracy for Arxiv (See Appendix E.3 for others). This trend suggests that LLMs benefit from greater domain-specific exposure during training or inference. However, some variation persists, likely influenced by factors such as terminology ambiguity, data formatting, or documentation consistency. These findings underscore the importance of high-quality domain data in enhancing LLM reliability for specialized industrial tasks.

| Asset Type | Total Q. | % Correct |
|---|---|---|
| Electric Motor | 234 | 43.59% |
| Steam Turbine | 171 | 47.95% |
| Aero Gas Turbine | 336 | 50.89% |
| Compressor | 220 | 56.36% |
| Power Transformer | 544 | 57.35% |
| Fan | 200 | 58.00% |
| Pump | 152 | 58.55% |
| Reciprocating Internal Combustion Engine | 336 | 65.48% |
| Industrial Gas Turbine | 240 | 70.83% |
| Electric Generator | 234 | 70.94% |

Table 5: Performance Analysis of Correct Answer Percentage and Total Questions for Each Asset Type

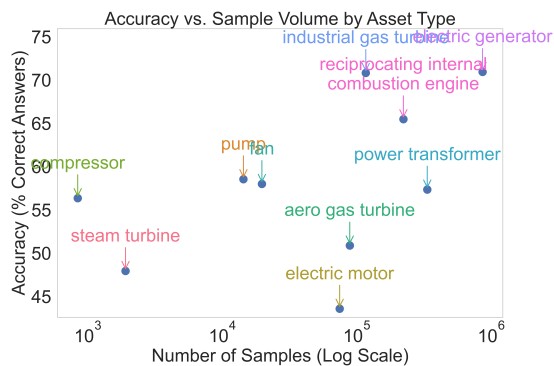

Figure 6: Accuracy vs. Sample Volume by Asset Type

## 4  Experimental Results

This section presents extended evaluation experiments on `SC-MCQA`, examining how reasoning-based prompting and external knowledge integration affect model performance. We further assess FailureSensorIQ through human expert evaluation to validate the quality of dataset.

**Impact of Reasoning-Based Prompting.** As shown in Figure 4, reasoning-centric models dominate leaderboard positions. To assess the impact of explicit reasoning, we evaluate llama models using different reasoning strategies: Chain-of-Thought (`CoT`) prompting in Standard, Expert, and Inductive configurations, and the `Plan@Solve` method (see Appendix E.2 for prompt templates and additional results). While `CoT@Standard` and `Plan@Solve` are general-purpose, the other two prompts are tailored for industrial contexts. As shown in Table 6, `CoT` prompting enables smaller models to match or even outperform larger models using direct prompting (see Baseline row). Interestingly, `llama-3-70b` improves from 41.69% to 51.18% with `CoT`. Moreover, accuracy scales with model size: the largest model, `4-Maverick-17B-128E`, achieves 56.90% average accuracy which is about **+41%** higher than the smallest model, `3.1-8b` (40.27%). Overall, reasoning-oriented prompts like `CoT@Standard` and `Plan@Solve` consistently enhance model performance across scales.

| Method/Model | 4-Mav-17B-128E | 4-scout-17b-16e | 3-405b | 3.3-70b | 3.1-8b |
|---|---|---|---|---|---|
| **Direct Prompt** | 55.83 | 53.96 | 51.26 | 41.69 | 40.04 |
| **COT@Inductive** | 56.88 | 54.22 | 53.17 | 49.46 | 40.27 |
| **COT@Expert** | 56.96 | 53.06 | 55.38 | 50.96 | 42.11 |
| **COT@Standard** | **57.29** | 52.53 | **55.57** | **51.18** | 45.74 |
| **Plan@Solve** | 56.47 | **55.46** | 54.89 | 50.36 | **46.46** |
| **Average** | 56.90 | 53.82 | 54.75 | 50.49 | 43.65 |
| **Baseline** | 60.40 o1 | 55.83 4-Maverick | 55.83 4-Maverick | 51.26 3-405b | 41.69 3.3-70b |

Table 6: Performance Comparison across Different Reasoning Models for llama models

**AI Agent with External Knowledgebase.** Given the sheer volume of content about industrial assets (as shown in Figure 6), we deploy a ReAct [46] agent equipped with Wikipedia and arXiv search tools to dynamically retrieve relevant information while answering questions. On average, the agent issues 2–3 queries per question, using both tools in an interleaved manner. However, our analysis reveals

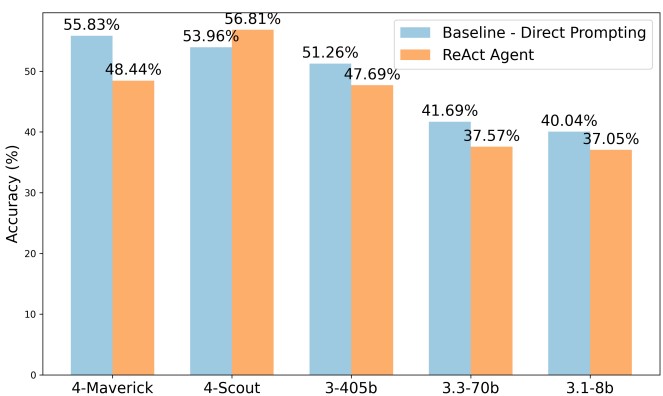

Figure 7: Accuracy comparison between Direct Prompting and ReAct across llama models.

Table 7: Expert Evaluation (Quiz Scores). Numbers in parentheses indicate "Don't know" responses.

| ID (Role) | Quiz 1 (%) | Quiz 2 (%) |
|---|---|---|
| A (Expert) | 55.56 (4) | 64.86 (3) |
| B (Expert) | 66.67 (1) | 65.71 (5) |
| D (Expert) | 57.89 (2) | 50.00 (0) |
| E (Expert) | – (– ) | 61.11 (22) |
| C (Practitioner) | 20.59 (6) | 43.24 (3) |

that such tool-augmented setups do not reliably enhance performance. Among five llama models, only one shows marginal improvement, while others including `llama-4-maverick` experience drops (e.g., from 55.83% to 48.44%). This suggests that success on industrial QA tasks depends more on reasoning over internal representations and navigating fine-grained distractors than on retrieving surface-level facts. These results highlight that just access to external knowledge, but the *method* of search and reasoning is a critical and currently underexplored dimension for tool-augmented agents in specialized domains.

**Human Evaluation**. We conducted a human study involving 5 participants, where 4 have expertise in industrial assets (reliability engineers) and 1 industrial data scientist. We selected a balanced sample of 80 questions spread across 10 asset types. Questions were balanced along two axes: query type (elimination vs. selection) and axis of reasoning (failure mode $\rightarrow$ sensor, and sensor $\rightarrow$ failure mode). We divided the 80 questions into 2 quizzes: Quiz 1 focused on elimination (select the *irrelevant* option), and Quiz 2 on selection (select the *most relevant* option). Since participants had varying expertise across asset types, a "don't know" option was included. Our setup is consistent with human evaluation methods in similar works (see Appendix J).

Table 7 shows the results. Domain expertise significantly improves performance on reasoning-intensive QA tasks in industrial contexts, with experts outperforming the practitioner by over 25 percentage points on average, and up to 39.45 points in Quiz 1. Inter-rater agreement among experts with IDs A, B and D yield a moderate Cohen's Kappa ($\kappa = 0.462$ overall [6]), indicating that while domain knowledge enhances accuracy, the task retains nuanced complexity. This suggests that the task is neither trivially objective nor entirely subjective, involving a diverse set of assets and underscoring its nuanced reasoning demands. These findings highlight the value of integrating expert knowledge into both QA system development and evaluation.

# 5 Multi-Correct MCQA

We report results for ten LLMs evaluated under the `MC-MCQA-Direct` protocol (Table 8), where each question requires selecting exactly two correct answers from a candidate set. The evaluation emphasizes both exact match and soft matching metrics such as F1, Jaccard, and a consistency-weighted score that penalizes partial but incorrect selections. Despite decent F1 scores (0.59), exact match remains under 21%, highlighting the challenge of jointly identifying all correct options. Models

like o4-mini and gpt-4.1-nano show relatively better consistency, but overall performance suggests that exact selection of multiple true answers remains a difficult task, especially without explicit guidance on how many options are correct.

Table 8: Performance on multi-correct MCQA (2-answer) benchmark using the `MC-MCQA` approach. EM = exact match.

| Model | EM | Precision | Recall | Micro F1 | Macro F1 | Hamming Loss | Set Size |
|---|---|---|---|---|---|---|---|
| o3 | 0.200 | 0.591 | 0.710 | 0.645 | 0.645 | 0.313 | 2.40 |
| o4-mini | 0.201 | 0.590 | 0.710 | 0.645 | 0.644 | 0.313 | 2.41 |
| gpt-4.1 | 0.186 | 0.590 | 0.676 | 0.630 | 0.630 | 0.317 | 2.41 |
| gpt-4.1-mini | 0.181 | 0.580 | 0.682 | 0.627 | 0.626 | 0.325 | 2.41 |
| gpt-4.1-nano | 0.186 | 0.586 | 0.682 | 0.630 | 0.630 | 0.320 | 2.41 |
| llama-4-maverick | 0.184 | 0.590 | 0.671 | 0.628 | 0.627 | 0.318 | 1.80 |
| llama-4-scout | 0.205 | 0.607 | 0.684 | 0.643 | 0.643 | 0.303 | 1.94 |
| llama-3-405b | 0.196 | 0.599 | 0.686 | 0.640 | 0.640 | 0.309 | 2.40 |
| llama-3-70b | 0.185 | 0.585 | 0.679 | 0.629 | 0.628 | 0.321 | 2.55 |
| llama-3-8b | 0.178 | 0.577 | 0.676 | 0.623 | 0.623 | 0.328 | 2.56 |

# 6 LLMFeatureSelect: scikit-learn Transformer

We implement `LLMFeatureSelector`, a prompt-based method that leverages an LLM to recommend relevant features given a problem statement, sensor names, and a target variable. As shown in the detailed system prompt in Appendix Figure 17, we use a one-step process to extract the important variables along with reasoning. While prior work has explored similar tasks [12], our goal is to evaluate LLM capabilities specifically in the industrial domain. We test the approach on three open-source, real-world datasets and assess performance by computing the correlation between the LLM-recommended features and the target variable (next timestep).

As seen in Table 9, 3 out of 6 test cases, the most highly correlated signal, which is highlighted, is among the LLM's top-5 suggestions, indicating promising alignment between model predictions and empirical sensor importance. Dataset descriptions and further findings are provided in Appendix L and correlation results in Table 9. `FailureSensorIQ` can complement this approach, as reasoning is central to effective feature selection.

Table 9: Absolute value Correlations between the top-5 recommended sensors and the target variable

| Asset | Task: (Failure or Energy Pred.) | Top-5 recommended features | | | | | Max Corr. |
|---|---|---|---|---|---|---|---|
| | | 1 | 2 | 3 | 4 | 5 | |
| Electrical Transf. | Magnetic Oil Gauge | 7.83 | 12.52 | 20.33 | 0.25 | 0.31 | 35.88 |
| Air Compressor | Bearings | 0.07 | 1.94 | 4.51 | 2.75 | 0.01 | 36.03 |
| Air Compressor | Water Pump | 15.28 | 21.38 | 13.62 | 15.87 | 14.52 | 21.38 |
| Air Compressor | Radiator | 31.85 | 31.83 | 31.78 | 86.88 | 25.16 | 86.88 |
| Air Compressor | Valve | 1.66 | 52.64 | 14.42 | 1.43 | 0.30 | 52.64 |
| Wind Mill | Energy Production | 92.02 | 82.79 | 82.83 | 36.0 | 35.75 | 94.40 |

# 7 Limitations and Future Work

A key limitation of FailureSensorIQ is its current focus on static knowledge, without modeling temporal sensor-failure dynamics. Additionally, performance varies due to uneven online knowledge availability across assets, affecting benchmarking consistency. Our experimental results underscore these limitations and present an opportunity for innovation in reasoning-aware and agentic LLM systems tailored to industrial diagnostics. Future work will extend the benchmark with temporal reasoning tasks to evaluate how well LLMs integrate dynamic signals into feature selection. Another direction is to expand the scope of the dataset by incorporating more assets and failure modes. Based on preliminary studies, synthetic data generation and knowledge distillation from bigger and more capable models to smaller ones are promising directions [17].

## Acknowledgments

The authors thank Sal Rosato, Muhammad Paracha and Debajyoti Chakraborty for their valuable help on this work.

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

# A   Dataset Benefits for AI and Industrial Research

Our dataset provides a range of opportunities across various domains. Below, we highlight seven key application areas: the first four focus on reasoning and knowledge integration ability, while the remaining three are centered around domain-specific applications.

- **🧠 MCQA Reasoning Analysis:** Focuses on LLMs' ability to solve multi-choice problems effectively [4, 29, 50, 51, 51].

- **🔍 ReAct-style Reasoning with Multiple Data Sources:** Supports dynamic retrieval of external knowledge sources (e.g., Wikipedia, ArXiv) for open-book experimentation [2, 46]

- **🔀 Knowledge Transfer Across Model Scales:** Explores model scalability, pushing the limits of smaller models compared to larger ones.

- **✅ Knowledge-Invariant LLM Testing:** Evaluates LLMs' consistency in applying domain-specific knowledge [14].

- **🗄 Domain-Specific Embedding Models:** Facilitates the creation of domain-specific embedding models for improved performance [16, 42].

- **⚙ Synthetic Data Generation for Domains:** Enables large-scale synthetic datasets for data-scarce industries [41].

- **📶 IoT Integration and Real-World Validation:** Validates AI models using real-world IoT data for continuous improvement [10, 44].

# B   Dataset Preparation : Real and Perturbed

We systematically create a dataset of multiple-choice questions (MCQs) using the relevant and irrelevant relationships between sensors and failure modes of an asset ($D_{relevant}$) which are found from expert curated tables (example in Table 1) and question templates ($QT_{fail}$ and $QT_{sensor}$). For each entry in $D_{relevant}$ it selects an appropriate question template, replacing placeholders with corresponding attributes like failure mode, sensor type, or asset class. We then identify the correct and incorrect answer pairs from the item's multiple-choice targets ($mc\_targets$), ensuring at least one correct answer $A$ and one incorrect answer $I$ is available ($|A| \geq 1$ and $|I| \geq 1$) to form meaningful questions. For each valid pair of correct answers, incorrect answers are sampled, shuffled, and integrated into a final MCQ format. The resulting questions with associated passage, answers, and indices of correct answers, are appended to the results list ($res_{relevant}$) which is returned as the output.

**Algorithm 1: Generate Multiple-Choice Questions**

**Input:** $D_{relevant}, QT_{fail}, QT_{sensor}, max\_n\_choices$
**Output:** $res_{relevant}$
Initialize $res_{relevant} \leftarrow [\,]$;
**for** *each item* $\in D_{relevant}$ **do**
    **if** $'failure\_mode' \in item$ **then**
        $q \leftarrow$ random.choice($QT_{fail}$);
        Replace placeholder in $q$ with $item["failure\_mode"]$;
    **else**
        $q \leftarrow$ random.choice($QT_{sensor}$);
        Replace placeholder in $q$ with $item["sensor"]$;
    Replace placeholder with $item["asset\_class"]$;
    Extract $qa\_pairs \leftarrow \{(key, value) \mid key \in item["mc\_targets"]\}$;
    Identify correct answers $A$, incorrect answers $I$;
    **if** $|I| < 1 \; or \; |A| < 1$ **then**
        **continue**;
    **for** *each pair* $(i, j) \in A$ **do**
        $correct \leftarrow [qa\_pairs[i], qa\_pairs[j]]$;
        Sample $incorrect \leftarrow$ random.sample($I, n$);
        Shuffle answers, find indices $A_{idx}$;
        $q_{final} \leftarrow$ Question($passage, answers, A_{idx}$);
        Append $q_{final}$ to $res_{relevant}$;

**return** $res_{relevant}$;

## B.1 Question Templates

We have developed question templates as shown in Table 10. We only show one example per task. The table provides structured templates for generating questions aimed at exploring the relationships between sensors, failure modes, and industrial asset classes, supporting tasks like Condition-Based Monitoring (CBM) and reliability analysis. Each task targets a specific aspect of relevance. FM2Sensor-Sel. focuses on identifying the most relevant sensors for a failure mode in an asset class, while FM2Sensor-El. identifies sensors that are not relevant. Similarly, Sensor2FM-Sel. determines the most relevant failure modes associated with abnormal sensor readings, and Sensor2FM-El. identifies failure modes that are irrelevant in such contexts. These templates enable systematic question formulation, facilitating analysis of sensor-failure mode interactions to enhance asset performance and reliability.

| Task | Question Template |
|------|-------------------|
| FM2Sensor-Sel. | For {asset_class}, if a failure event {failure_mode} occurs, which sensors out of the choices are the most relevant regarding the occurrence of the failure event? |
| FM2Sensor-El. | For {asset_class}, if a failure event {failure_mode} occurs, which sensors out of the choices are not relevant regarding the occurrence of the failure event? |
| Sensor2FM-Sel. | In the context of {asset_class}, which failure modes are the most relevant when {sensor} shows abnormal readings? |
| Sensor2FM-El. | In the context of {asset_class}, which failure events are not relevant when the sensor {sensor} shows an abnormal reading? |

Table 10: Examples of question templates for sensor selection and failure mode identification. Each template is rephrased multiple times for different tasks.

## B.2 Raw Data

We have 10 assets at present in our analysis, and per asset we have two mapping information available in a form of Knowledge Graph (KG). Figure 8 provide a KG view.

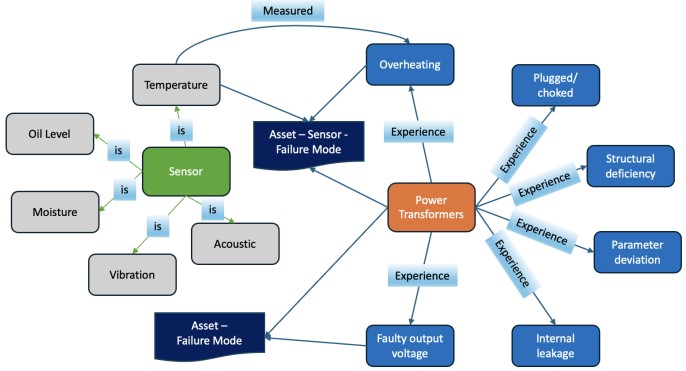

Figure 8: Knowledge Graph Interaction

## B.3 Perturbed Pipeline

The following Figure 9 show a pipeline for data generation. The pipeline depicted in the diagram represents a systematic process for perturbing questions through a series of transformations. It starts with the Original Question (Q) and progresses through multiple stages, each applying a specific modification. The first stage, OptionCaesar, introduces a basic transformation to the options, such as encoding or shifting values. Next, OptionPerm rearranges the order of the options without changing their content. The OptionForm stage modifies the format of the options, such as rephrasing or altering their presentation. Following this, ChangeType alters the question type, for example, converting a multiple-choice question to a true/false format. The pipeline then proceeds to Question Paraphrase, where the question itself is rephrased while preserving its original meaning. Finally, the SwapPos stage changes the positions of key elements within the question, such as swapping the subject and predicate.

The pipeline produces two potential outputs: an Easy Perturbed Question (Q) after moderate transformations (indicated by a green arrow) or a Hard Perturbed Question (Q) after more extensive

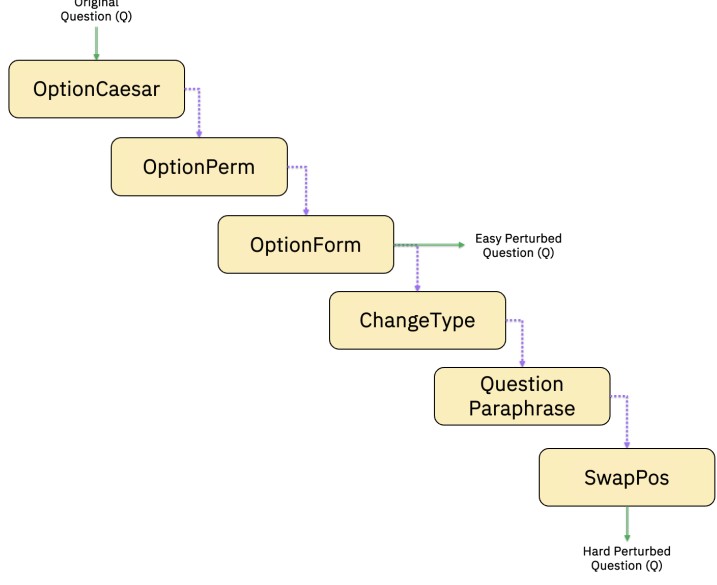

Figure 9: Pipeline for Preparing Perturb Dataset

transformations at the final stage. The dotted arrows between stages represent the sequential application of these perturbation techniques, progressively increasing the complexity of the modifications.

## C  Experimental Configuration

We use IBM watsonx and Microsoft Azure platforms which host various LLMs to perform the inference for the different experiments. For models that are not available on these platforms, we run on an internal GPU cluster equipped with 8 × Nvidia A100 (80 GB) and 16 × AMD EPYC 7763 64-Core CPUs, along with 128 GB RAM and 1TB SSD.

## D  Result Trace Analysis Demonstration

For the input prompt shown in Figure 10, we provide the reasoning and answers for o1, llama-3.3-70b, and mistral-large-instruct-2407. We're given the failure mode "fuel filter blockage" for the asset "reciprocating internal combustion engine" it is asked to find the most relevant sensors. OpenAI o1 (Figure 11) constructs a causal chain based on the effects of blockage in the fuel filter. It reasons that it "reduces the amount of fuel filter reaching the engine cylinders". This leads to "incomplete combustion or a lean fuel mixture". Then this causes "lower combustion pressure within cylinders". Based on this causal analysis the model now selects the options: (C) Cylinder Pressure Sensor and (E) Output Power Sensor. However, the correct answer is (B) Exhaust Pressure.

In similar lines, Mistral-Large-Instruct (Figure 12) forms a reasoning in the form of a causal chain and comes with the answers (B) Exhaust Pressure and (E) Output Power. It gets 1 out of the 2 correct.

In contrast with the previous 2 models, Llama-3.3-70B-Instruct (Figure 13) does not go in the same depth as the other models. We see that it enumerates the options and just decides whether the sensors are related or not with minimal reasoning. It finally selects (C) Cylinder Pressure which is incorrect.

The example presented in this section highlights the importance of the deep reasoning to be able to answer these questions.

---

**Input Prompt**

Please select the correct option(s) from the following options given the question.
To solve the problem, follow the **"Let me think step by step reasoning strategy"**.
**Question:** Which sensor out of the choices can indicate the presence of **fuel filter blockage** in asset **reciprocating internal combustion engine**?
**Options**:
A air flow
B exhaust pressure
C cylinder pressure
D engine temperature
E output power
**Your output must strictly follow this format:**
{"reasoning": <"Your reasoning step-by-step">, "answer": <the list of selected options, e.g., ["A", "B", "C", "D", "E"]>}
**Your output:**

---

Figure 10: Input Prompt. Ground truth is B, Exhaust Pressure.

**OpenAI o1 Reasoning and Response**

**Reasoning:**
A blockage in the fuel filter reduces the amount of fuel reaching the engine cylinders. This leads to incomplete combustion or a lean fuel mixture, causing lower combustion pressure within the cylinders.
Therefore, a **cylinder pressure sensor (Option C)** can detect reduced pressures indicating a fuel filter blockage.
Additionally, with less fuel available for combustion, the engine's performance declines, resulting in decreased power output.
Hence, an **output power sensor (Option E)** can also indicate a drop in engine power due to fuel filter blockage.
Other sensors like air flow, exhaust pressure, and engine temperature are less directly affected or may not provide a clear indication of fuel filter blockage.
**Answer:**
C, E

Figure 11: Reasoning Trace by o1.

**Mistral-Large-Instruct-2407 Reasoning and Response**

**Reasoning:**
Let me think step by step:
1. A fuel filter blockage would restrict the flow of fuel to the engine.
2. This restriction would lead to a reduction in the engine's ability to burn fuel efficiently.
3. Inefficient fuel burning would result in a decrease in output power and an increase in exhaust pressure due to incomplete combustion.
4. Other sensors like air flow, cylinder pressure, and engine temperature might not directly indicate a fuel filter blockage as they are influenced by various factors.

Therefore, the most indicative sensors for a fuel filter blockage would be exhaust pressure and output power.
**Answer:**
B, E

Figure 12: Reasoning Trace by Mistral-Large-Instruct-2407.

**Llama-3.3-70B-Instruct Reasoning and Response**

**Reasoning:**
To determine the presence of a fuel filter blockage in an asset reciprocating internal combustion engine, we need to consider the effects of such a blockage on the engine's operation. A fuel filter blockage would restrict the flow of fuel to the engine, leading to a decrease in engine performance and potentially causing the engine to run rich or lean.
Let's analyze the options:
A. Air flow: A fuel filter blockage would not directly affect air flow, so this option is unlikely to indicate the presence of a blockage.
B. Exhaust pressure: While a fuel filter blockage might affect engine performance, it would not directly impact exhaust pressure, making this option less likely.
C. Cylinder pressure: A fuel filter blockage would lead to a decrease in fuel flow, resulting in lower cylinder pressures. This option is a strong candidate.
D. Engine temperature: A fuel filter blockage might cause the engine to run rich or lean, potentially affecting engine temperature. However, this is an indirect effect, and engine temperature is not the most reliable indicator of a fuel filter blockage.
E. Output power: A fuel filter blockage would likely cause a decrease in engine output power, but this is a more general indicator of engine performance issues and not specific to fuel filter blockages.
Considering the options, the most direct and reliable indicator of a fuel filter blockage is a decrease in cylinder pressure
**Answer:**
C

Figure 13: Reasoning Trace by Llama-3.3-70B.

# E  Additional Experimental Results

## E.1  Leaderboard

In this section, we present the performance of 24 different LLMs on the `SC-MCQA` and its complex perturbation version `ComplexPert`.

| Model | Acc. Total | Acc. Sel. | Acc El. | Acc Perturb. | Acc. Consist. |
|---|---|---|---|---|---|
| o1 | 60.40 | 61.06 | 67.89 | 49.76 | 44.17 |
| o3-mini | 58.46 | 56.81 | 69.84 | 53.28 | 47.47 |
| llama-4-mav.-17b-128e | 55.83 | 44.47 | 71.90 | 49.12 | 40.83 |
| llama-4-scout-17b-16e | 53.96 | 44.47 | 63.53 | 29.36 | 24.11 |
| gpt-4.1 | 53.47 | 56.38 | 59.17 | 45.74 | 39.93 |
| o1-preview | 52.31 | 49.57 | 62.5 | 47.51 | 40.68 |
| llama-3.1-405b-instruct | 51.26 | 48.72 | 61.24 | 40.04 | 30.93 |
| ds-r1 | 50.09 | 45.74 | 59.75 | 54.37 | 42.93 |
| mistral-large-instr.-2407 | 50.09 | 51.28 | 57.57 | 38.10 | 29.32 |
| gpt-4.1-mini | 49.27 | 45.53 | 57.34 | 39.97 | 31.76 |
| phi-4 | 48.56 | 40.43 | 60.32 | 36.30 | 29.62 |
| mixtral-8x22b-v0.1 | 45.18 | 42.55 | 59.06 | 27.52 | 19.57 |
| ds-r1-dist.-llama-7b | 44.62 | 36.38 | 65.14 | 44.99 | 30.30 |
| gemma-2-9b | 43.98 | 30.43 | 58.6 | 45.29 | 36.07 |
| ds-r1-dist.-llama-8b | 43.04 | 38.94 | 54.36 | 24.26 | 16.27 |
| gpt-4.1-nano | 41.77 | 40.00 | 50.34 | 14.47 | 9.30 |
| llama-3.3-70b-instruct | 41.69 | 35.11 | 55.85 | 37.57 | 25.27 |
| llama-3.1-8b-instruct | 40.04 | 36.17 | 51.15 | 25.35 | 13.95 |
| llama-3.2-11b-vision | 39.11 | 33.83 | 50.92 | 45.74 | 30.90 |
| qwen2.5-7b-instruct | 38.73 | 40.64 | 49.54 | 28.20 | 16.42 |
| ds-r1-dist.-qwen-7b | 34.01 | 23.83 | 50.11 | 17.02 | 8.02 |
| granite-3.2-8b-instruct | 30.26 | 41.70 | 29.82 | 19.24 | 8.74 |
| mixtral-8x7b-v0.1 | 27.60 | 25.32 | 38.19 | 11.21 | 4.46 |
| granite-3.3-8b-instruct | 25.83 | 32.13 | 29.47 | 23.73 | 14.40 |
| granite-3.0-8b-instruct | 22.85 | 16.17 | 36.70 | 16.16 | 4.87 |

Table 11: Performance comparison of different models across total, selection, elimination, perturbation, and consistency accuracies.

| Model | Electric Motor | Steam Turbine | Aero Gas Turbine | Industr. Gas Turbine | Pump | Compressor |
|---|---|---|---|---|---|---|
| o1 | 68.8 | 47.95 | 50.89 | 70.83 | 58.55 | 56.36 |
| o3-mini | 63.25 | 49.71 | 51.49 | 68.33 | 52.63 | 55.45 |
| llama-4-maverick-17b-128e | 66.24 | 47.37 | 47.32 | 69.17 | 55.26 | 52.27 |
| llama-4-scout-17b-16e | 60.26 | 40.94 | 47.02 | 67.5 | 50.0 | 46.36 |
| gpt-4.1 | 58.12 | 43.27 | 43.75 | 69.17 | 52.63 | 46.36 |
| o1-preview | 65.38 | 43.86 | 42.56 | 62.92 | 50.0 | 46.82 |
| llama-3.1-405b-instruct | 61.11 | 39.18 | 48.51 | 59.17 | 46.05 | 45.0 |
| mistral-large-instruct-2407 | 51.71 | 36.26 | 44.64 | 59.58 | 45.39 | 46.82 |
| deepseek-r1 | 58.97 | 39.18 | 44.94 | 63.33 | 50.66 | 44.09 |
| gpt-4.1-mini | 55.98 | 43.27 | 41.67 | 56.67 | 45.39 | 43.18 |
| phi-4 | 50.85 | 40.35 | 45.83 | 57.92 | 50.0 | 41.36 |
| mixtral-8x22b-v0.1 | 46.58 | 40.35 | 39.29 | 57.5 | 46.05 | 43.18 |
| deepseek-r1-distill-llama-70b | 50.0 | 36.84 | 40.77 | 51.25 | 39.47 | 34.55 |
| gemma-2-9b | 45.3 | 35.67 | 42.86 | 53.33 | 43.42 | 40.91 |
| deepseek-r1-distill-llama-8b | 46.58 | 36.26 | 43.45 | 50.0 | 40.79 | 41.82 |
| gpt-4.1-nano | 44.87 | 31.58 | 38.99 | 50.83 | 40.13 | 37.73 |
| llama-3.3-70b-instruct | 45.3 | 29.24 | 35.42 | 45.42 | 37.5 | 35.91 |
| llama-3.1-8b-instruct | 39.74 | 33.33 | 37.5 | 48.75 | 36.84 | 35.91 |
| llama-3.2-11b-vision | 38.46 | 31.58 | 36.61 | 42.92 | 38.16 | 37.73 |
| qwen2.5-7b-instruct | 41.03 | 30.41 | 35.42 | 45.0 | 39.47 | 35.0 |
| deepseek-r1-distill-qwen-7b | 37.18 | 28.07 | 32.44 | 43.33 | 34.87 | 31.82 |
| granite-3.2-8b-instruct | 34.19 | 27.49 | 24.7 | 30.0 | 35.53 | 28.18 |
| mixtral-8x7b-v0.1 | 29.06 | 20.47 | 20.24 | 24.58 | 23.68 | 22.73 |
| granite-3.3-8b-instruct | 26.5 | 24.56 | 19.05 | 26.67 | 28.95 | 26.36 |
| granite-3.0-8b-instruct | 28.63 | 19.3 | 18.75 | 19.17 | 23.68 | 20.45 |

Table 12: FailureSensorIQ performance per asset (Part A).

| Model | Recipr. Intern. Comb. Engine | Electric Generator | Fan | Power Transformer |
|---|---|---|---|---|
| o1 | 65.48 | 70.94 | 58.0 | 57.35 |
| o3-mini | 63.39 | 65.81 | 61.0 | 54.78 |
| llama-4-maverick-17b-128e | 61.01 | 62.39 | 56.5 | 48.71 |
| llama-4-scout-17b-16e | 66.96 | 59.4 | 60.5 | 45.04 |
| gpt-4.1 | 61.01 | 58.12 | 57.0 | 48.9 |
| o1-preview | 58.33 | 62.82 | 53.5 | 44.85 |
| llama-3.1-405b-instruct | 58.63 | 55.56 | 51.5 | 46.51 |
| mistral-large-instruct-2407 | 59.52 | 50.85 | 50.0 | 49.45 |
| deepseek-r1 | 51.49 | 56.84 | 53.5 | 44.3 |
| gpt-4.1-mini | 56.85 | 49.15 | 54.5 | 46.69 |
| phi-4 | 52.98 | 52.56 | 55.0 | 43.38 |
| mixtral-8x22b-v0.1 | 52.38 | 39.74 | 53.5 | 39.71 |
| deepseek-r1-distill-llama-70b | 52.98 | 49.15 | 48.5 | 41.18 |
| gemma-2-9b | 53.57 | 44.44 | 55.0 | 33.82 |
| deepseek-r1-distill-llama-8b | 55.65 | 52.14 | 43.5 | 29.6 |
| gpt-4.1-nano | 50.89 | 45.73 | 49.5 | 33.27 |
| llama-3.3-70b-instruct | 50.0 | 42.74 | 48.0 | 41.91 |
| llama-3.1-8b-instruct | 47.92 | 37.61 | 46.5 | 36.4 |
| llama-3.2-11b-vision | 47.62 | 36.32 | 48.5 | 34.93 |
| qwen2.5-7b-instruct | 47.62 | 42.74 | 44.5 | 31.62 |
| deepseek-r1-distill-qwen-7b | 41.67 | 35.04 | 34.0 | 26.84 |
| granite-3.2-8b-instruct | 36.9 | 28.63 | 33.0 | 27.94 |
| mixtral-8x7b-v0.1 | 36.61 | 23.08 | 34.0 | 32.17 |
| granite-3.3-8b-instruct | 36.31 | 25.21 | 30.5 | 20.77 |
| granite-3.0-8b-instruct | 32.14 | 22.22 | 27.0 | 19.12 |

Table 13: FailureSensorIQ performance per asset (Part B).

## E.2  Trigger Statement for Induced Reasoning

Table 14 shows the trigger statement we have used as a part of CoT based prompting to LLM. Similarly, Table 15 shows the prompt for Plan-Solve mechanism.

| CoT Stype | Trigger Statement |
|---|---|
| Standard | Let me think step by step |
| Expert | Let me think step by step as a reliability engineer |
| Inductive | Let's use step by step inductive reasoning, given the domain specific nature of the question |

Table 14: CoT Style

| Plan-Solve | Trigger Statement |
|---|---|
| prompt-3 | Let's first prepare relevant information and make a plan.  Then, let's answer the question step by step (pay attention to commonsense and logical coherence). |

Table 15: Plan-Solve

## E.3  Probing External Knowledge Sources

In this section, we examine how external knowledge sources are probed to estimate the volume of documents they possess, which may have been leveraged during model fine-tuning. To operationalize this, we use keyword-based querying across multiple platforms (Wikipedia, arXiv, CrossRef, and Google) and record the number of matching documents related to each asset type.

We document the querying process using a sample pseudocode procedure in Algorithm 2, which outlines how we retrieved the number of Wikipedia entries associated with a given keyword. Similar script is developed for the other sources. Our goal is to understand the potential influence of these sources on the model's performance by analyzing the extent and nature of available data related to specific asset types.

---

**Algorithm 2:** Search Wikipedia for Keyword Hits

**Input:** Search keyword $k$
**Output:** Total number of matching pages $h$
Define Wikipedia API endpoint: `url ← "https://en.wikipedia.org/w/api.php"` ;
Construct query parameters: `params ← { action: "query", list: "search", srsearch:` $k$,
 `format: "json" }` ;
Send HTTP GET request: `response ← GET(url, params)` ;
Parse JSON response: `data ← ParseJSON(response)` ;
Extract hit count: $h$ ← `data["query"]["searchinfo"]["totalhits"]` ;
**return** $h$ ;

---

The distribution of document volume shown in Figures: 14-16 reveals discrepancies across sources:

- **CrossRef Scholarly Dominance**: Assets like *industrial gas turbine*, *power transformer*, and *reciprocating internal combustion engine* exhibit extremely high coverage in CrossRef (e.g., over 1.8M entries for *industrial gas turbine*). This suggests substantial representation in peer-reviewed literature, reflecting their critical role in energy and infrastructure systems.

- **Wikipedia vs. arXiv Divergence**: While *fan* has extensive representation on Wikipedia (over 253K articles), it has modest coverage on arXiv (20K). Conversely, *electric motor* is strongly represented on arXiv (75K) but appears in fewer Wikipedia articles (33.5K). This contrast implies a split between assets emphasized in technical vs. general public domains.

- **Underrepresented Public Topics**: Assets like *aero gas turbine* and *electric generator* have very low Wikipedia visibility (fewer than 1.5K hits each), yet they appear prominently in arXiv and CrossRef. This suggests that these domains are underrepresented in public web knowledge despite being well-documented in scientific literature.

- **Public vs. Research Interest Misalignment**: *Pump* and *fan* have significant Google presence (1.4B and 7.08B hits, respectively), underscoring their ubiquitous usage. However,

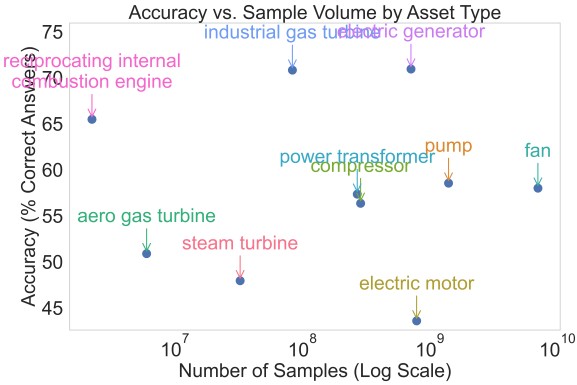

Figure 14: Google: Accuracy vs. Sample Volume

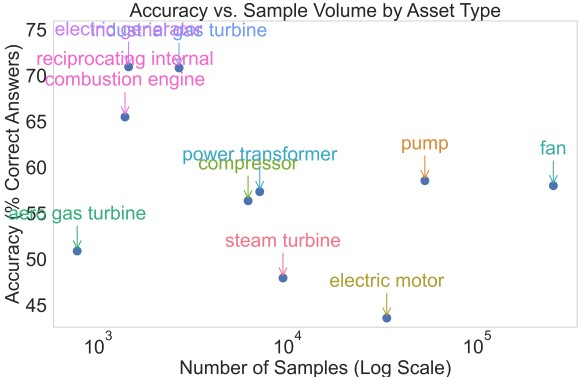

Figure 15: Wikipedia: Accuracy vs. Sample Volume

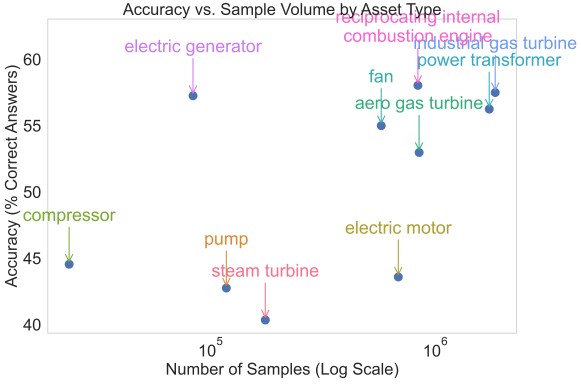

Figure 16: CrossRef: Accuracy vs. Sample Volume

their academic representation is comparatively limited, pointing to a potential mismatch between public visibility and scientific exploration.

- **Research Intensity Metric**: We compute a simple research intensity ratio, defined as CrossRef volume over arXiv volume, to assess where scholarly publication concentrates:
  - *Steam turbine* has the highest ratio (91.7), suggesting more journal-level than pre-print research.
  - *Electric generator* exhibits a low ratio (0.10), possibly indicating a heavier reliance on open-access pre-prints.

**Implications for Model Exposure and Bias**

These findings reveal a skewed exposure landscape across assets:

- Assets with higher web-scale and academic document volumes are more likely to have influenced pretrained LLMs.

- The absence or presence of certain assets across specific knowledge modalities (e.g., Wikipedia vs. CrossRef) may lead to knowledge blind spots or biases in LLM outputs.

Therefore, understanding these variations is critical for evaluating knowledge-dependent tasks, such as multiple-choice question answering in technical domains.

**Asset-Specific Challenges:**

- Asset Complexity: We can consider the difficulty of the questions in the asset level. Assets have a varying number of connected components and subsystems and their dynamics. For instance, Steam Turbine has many components and requires knowledge in thermodynamics, materials science, and rotor dynamics.

- Knowledge Gaps: Alternatively, is the complexity due to limited understanding of specific failure modes, sensor interactions, or component behaviors?

- Amount of reasoning needed to combine knowledge and infer relationships: Combining knowledge with logical deductions to answer. Example reasoning in Appendix Figures 11, 12, 13.

### E.4 Fine-tuning on HotpotQA

We conduct an experiment to demonstrate that the existing datasets lack information related to industrial assets to support the importance of FailureSensorIQ dataset to bridge this gap. We select HotpotQA [45] which is a dataset that consists of 113K question, answer pairs from Wikipedia and fine-tune Flan-T5-XL [25]. We ignore any contextual information and only fine-tune on the questions and answers and generate the answer conditioned on the question. A full fine-tuning is done for 3 epochs, using an A100 80GB GPU with a batch size of 16. The results reveal a drop in accuracy after fine-tuning from 36% to 29% which can be attributed to catastrophic forgetting, and the lack of information about industrial assets in HotpotQA. This result reinforces the importance for a domain-specific dataset for industrial assets like FailureSensorIQ.

### E.5 Analyzing Failure Mode Coverage from Online Industrial Datasets

We analyze 135 publicly available industrial datasets with many sourced from platforms like Kaggle and the UCI Machine Learning Repository [34]. The motivation is to evaluate them through the lens of failure mode coverage, as described in their dataset documentation or challenge descriptions.

Despite the diversity of assets and use cases, a common pattern emerges: most datasets focus on a single failure mode analysis, limiting their applicability in real-world, large-scale predictive maintenance systems. This highlights a scalability gap in the literature when it comes to handling multiple concurrent failure modes per asset.

**Analysis metrics:**

- Total datasets analyzed - 135

- Datasets with work order information - 1

- Datasets with time series sensor data - 84

- Datasets explicitly mentioning failure modes - 53
- Datasets with alert/alarm signals - 3
- Failure Mode Count Distribution - 0: 93, 1: 18, 2: 9, 3: 4, 4: 5, 5: 2, 6: 1, 7: 1, 8: 1, 10: 1.

**Key Findings:**
- From this distribution we conclude that 93 datasets do not mention any failure modes.
- Only 42 datasets describe more than one failure mode.
- Just 6 datasets include four or more failure modes.

This underscores a critical limitation in the existing datasets: limited support for multiple failure modes scenarios, which are common in operational industrial systems.

## F   Additional Dataset Formats

### F.1   True/False format

We prepare a True/False version similar to TruthfulQA [18], comprising 2,995 questions. For each question, we provide a statement describing the relationship between an asset's failure mode and a sensor and then ask whether it is true or false. For each asset and failure mode and sensors combination we construct a question. We run this dataset through the Perturbation and Uncertainty evaluation pipelines, as described in Section 3. Answers with invalid formatting are considered incorrect. We report results for several 8B models in Tables 16, 17.

**Perturbation Analysis (Similar to Table 4):**

| Model | ACC@Original | ACC@Simple | ACC@Complex | Consistency |
|---|---|---|---|---|
| Llama-3.1-8B-Instr | 49.30 | 47.07 | 42.52 | 27.38 |
| Qwen3-8B | 65.71 | 64.90 | 63.88 | 49.26 |
| Mistral-8B-Instr-2410 | 49.63 | 48.78 | 44.21 | 26.26 |
| Granite-3.3-instr | 58.81 | 34.57 | 52.64 | 21.52 |

Table 16: True/False question format Perturbation Analysis.

**Uncertainty Quantification (Similar to Table 3):**

| Model | UAcc | SS | CR | Acc |
|---|---|---|---|---|
| Llama-3.1-8B-Instr | 88.06 | 1.82 | 95.11 | 65.14 |
| Qwen3-8B | 47.44 | 1.89 | 93.03 | 36.57 |
| Mistral-8B-Instr-2410 | 88.53 | 1.79 | 95.17 | 63.91 |
| Granite-3.3-Instr | 81.75 | 1.80 | 94.56 | 59.57 |

Table 17: True/False question format Uncertainty Analysis.

Accuracy is naturally higher since there are only 2 options when compared to the original multiple-choice which mostly has 5 options. We observe a similar phenomenon of performance degradation with simple and complex perturbation of the prompt. Overall, performance on original > perturb simple > perturb complex, with only exception of granite which had severe performance degradation on perturbed complex questions.

Consistency is below 50%, and the Set Score Size (SS) is close to 2, despite the Uncertainty Accuracy (UAcc) being higher than 80%. This indicates that in the majority of cases, the model is highly uncertain between the two options. These results will be included in the Appendix, along with a mention of additional datasets as part of our key contributions. We encountered some difficulty with the Qwen3-8B model during the uncertainty evaluation. This is due to the <think> output format, and the fact that the underlying implementation does not yet support reasoning-based models.

### F.2   Open-Ended question format

**Question Prompt Generation**: We manually create 88 open-ended questions, such as: "List all failure modes of an electric motor that can be detected by vibration, cooling gas, or axial flux sensors". These open-ended questions can vary in complexity, especially when combining multiple aspects

such as different sensor types, as shown in the example or assets or failure modes. The answer to all the questions is in a list form and the average number of items in the ground truth set is 5.7.

**Evaluation**: Open ended question is complex problem than MCQA from generation and evaluation purpose. Such questions may yield a single answer or responses long enough to fill the entire context window. To address this, our system prompt (to be included as part of the paper) explicitly provides guidance: "Include only 1 to {item_count} items in a Python-style list, e.g., ["answer 1", "answer 2"]", where {item_count} is dynamically set to 5 or 10 based on the evaluation setting. We adopt a two-phase approach: a generation phase to generate potential candidate answers, followed by an extraction phase which structures the output into a valid JSON format. We then compare the generated list with the ground truth using structured semantic entity evaluation metrics for evaluation [20] and obtained precision and recall.

To assess the generation performance, we experiment with five models on this new set of open-ended questions. Results are reported in 18.

| Model | Precision@5 | Recall@5 | Precision@10 | Recall@10 |
|---|---|---|---|---|
| Ministral-8B-Instruct | 0.2877 | 0.0563 | 0.2144 | 0.0695 |
| mistral-medium-2505 | 0.3273 | 0.2545 | 0.1778 | 0.2305 |
| **mistral-large** | **0.3826** | **0.4089** | 0.2349 | **0.3945** |
| Llama-3.1-8B | 0.1955 | 0.0367 | 0.1216 | 0.1018 |
| Llama-3-3-70b | 0.3205 | 0.1624 | **0.2812** | 0.2022 |
| Llama-3-405b | 0.2554 | 0.2407 | 0.1933 | 0.2739 |
| Llama-4-maverick | 0.3251 | 0.2808 | 0.2579 | 0.2826 |
| granite-3-3-8b | 0.0732 | 0.0283 | 0.1086 | 0.0486 |
| Qwen-3-8B | 0.3229 | 0.1301 | 0.2292 | 0.0874 |

Table 18: Open-ended question format performance. Precision@K means that for each question there are K candidate answers.

Increasing the number of candidates from top-5 to top-10 did not significantly improve recall and consistently reduced precision. This indicates that most relevant results are already captured within the top-5, and additional candidates introduce noise. Mistral-large seems to have the best performance overall.

## G   Explicit mention of the Number of Correct Options

In our original experiments in Section E.1 we did not explicitly inform the models of the number of correct choices to align with real-world scenarios, where such information is not provided which is harder hence low exact match (EM) rates. We conduct an additional experiment with with select LLMs in which we explicitly mention the number of correct options where we observe that they perform better. We perform this on the Single Correct MCQA (SC-MCQA).

| Model | EM | F1 | Set Size |
|---|---|---|---|
| o3 | 0.384 | 0.635 | 2.0 |
| o4-mini | 0.382 | 0.633 | 2.0 |
| gpt-4.1 | 0.385 | 0.633 | 2.0 |
| gpt-4.1-mini | 0.385 | 0.635 | 2.0 |
| gpt-4.1-nano | 0.384 | 0.634 | 2.0 |
| llama-4-maverick | 0.434 | 0.678 | 2.0 |
| Llama-4-scout | 0.403 | 0.661 | 1.99 |
| llama-3-1-405b | 0.376 | 0.635 | 2.0 |
| llama-3-3-70b | 0.343 | 0.610 | 2.0 |
| Llama-3.1-8B | 0.252 | 0.561 | 2.16 |

Table 19: Model performance when number of correct options is provided.

# H    Does External Knowledge Help?

To investigate whether access to external tools enhances performance on MCQA, we evaluate a ReAct agent equipped with Wikipedia and arXiv search capabilities. The agent is allowed to autonomously choose sources, formulate search queries, and perform iterative reasoning to reach an answer.

Surprisingly, the tool-augmented setup does not outperform baseline prompting. In fact, for LLaMA-3-70B, the accuracy **drops from 41.7% (baseline) to 37.6% (ReAct)**. Across all five llaMA models, only one show marginal gains, while the remaining four experience clear declines. This suggests that answers to MCQA questions are generally **not directly retrievable** from external corpora such as Wikipedia or arXiv.

Instead, successful performance appears to require **reasoning over latent knowledge**, internal representations, and the ability to distinguish fine-grained distractors, which are capabilities not easily captured by Retrieval-Augmented Generation (RAG) alone. These findings underscore the benchmark's value as a test of internal reasoning, rather than surface-level factual recall.

We conduct an experiment where a ReAct agent was executed using Wikipedia and arXiv as external tools. Table 6 presents the results across several llama models. For each input question $Q$, the agent autonomously selects the source, issued queries, and performs iterative reasoning before generating an answer. The process is implemented using the LLaMA-3-70B model.

Surprisingly, the use of external tools did not yield consistent improvements. In fact, for most models, ReAct-based reasoning led to a drop in correct response rates compared to direct prompting. For instance, the `3.3-70b` model dropped from 41.69% (baseline) to 37.57%, and `3.1-8b` from 40.04% to 37.05%. Only one configuration, `4-scout-17b-16e`, showed marginal improvement. Additionally, ReAct executions introduced more invalid or malformed responses and incorrect answer formats, suggesting limitations in decision quality when navigating external sources.

These findings point to a broader implication: merely incorporating external information retrieval is insufficient for this task. The performance degradation under ReAct suggests that success depends not on accessing facts alone, but on executing multi-step reasoning that integrates and interprets information coherently. In that sense, this task poses a meaningful challenge beyond conventional retrieval-augmented generation (RAG) pipelines and instead aligns more closely with benchmarks that test deliberate, high-quality reasoning.

# I    Data Collection Details and Demographics

We leverage our internal reliability strategy library which has information for hundreds of assets and thousands of failure modes, to get the list of failure modes, and then our experts with a diverse background in mechanical and electrical engineering provide us the Failure Modes and Sensors mapping. The evaluators are experts in industrial maintenance products and have helped in assessing the dataset's correctness and difficulty and have an extensive knowledge about the assets' operations. There were eight male experts: two from Spain, two from United Kingdom, one from India, one from Brazil, one from Canada, and two from the United States. Everyone has at least twenty years of experience. The composition of our industrial product team is:

- Asset Performance Management (APM) – Focuses on building solutions for physical asset lifecycle management, composed of reliability engineers and electrical engineers.

- Operational Site Management – Site managers and plant operators with years of hands-on experience managing critical infrastructure in data center environments.

- Industrial AI Community – Industrial data scientists and SMEs involved in deploying AI solutions in domains such as Oil & Gas.

# J    Human Evaluation in Dataset Development

Human evaluation plays a pivotal role in the development of reliable and high-quality benchmark datasets, especially in domain-specific or knowledge-intensive contexts. Evaluators ranging from domain experts to informed non-experts contribute to critical tasks such as validating answer correctness, assessing question formulation, verifying distractor quality, and establishing human baselines. The experts are from our industrial product team and helped in assessing the dataset's correctness

and difficulty. The design and scope of these human studies significantly influence the dataset's robustness and the interpretability of model performance.

**Evaluator Expertise and Responsibility.** The choice of evaluators is often aligned with the dataset's domain complexity. For example, **StatQA** [52] engaged six postgraduate students (three with statistics backgrounds and three without) to evaluate 10% of the *mini-StatQA* dataset, comprising 117 examples. These evaluators participated in both closed-book and open-book assessments, enabling comparisons across expertise levels.

In **SPIQA** [47], a single experienced AI/ML researcher evaluated a 20% subset of the Test-A set. Given the dataset's cross-domain nature, the authors emphasized the challenge of obtaining a stable human performance upper bound and recommended future work include multiple evaluators from diverse scientific fields.

Similarly, **MEQA** [15] involved human verification of 100 questions to ensure answer correctness and question clarity, while **Wang et al.** [35] collected answers from three physicians for 120 medical questions, highlighting the need for expert-grounded validation in high-stakes domains.

**Multi-Phase Evaluation and Error Taxonomy.** The **MMLU-Pro** benchmark [37] implemented a two-phase validation pipeline. Phase 1 focused on correctness and appropriateness, removing problematic items such as proof-based, image-dependent, or poorly worded questions. In Phase 2, experts reviewed distractors to ensure that flagged options were clearly incorrect and distinguishable from the correct answer. This procedure identified and corrected systematic issues, including false negatives and unsuitable formats.

**Scale, Consistency, and Annotation Design.** Ying et al. [48] sampled 120 data points and employed five senior computational linguistics researchers to rate questions on fluency, coherence, and category accuracy, and answers on factuality, coherence, and correctness. Results included full score rates and inter-annotator agreement, underscoring the importance of evaluation consistency.

Other recent efforts, such as **BlendBench** [21], emphasize expert review of question quality, and **CiteMe** [26] adopts conventional human validation strategies to refine citation correctness. **Wen et al.** [40] reviewed 200 instruction examples to ensure clarity and task relevance.

**Summary.** Across benchmarks, human evaluation typically covers 10–20% of dataset samples, though methodologies and evaluator roles vary significantly. Some focus on answering questions (e.g., StatQA, Wang et al.), while others emphasize correctness verification, coherence, and distractor evaluation (e.g., MMLU-Pro, Automating Dataset Updates). These subsets, often ranging between 100–200 examples, are generally sufficient to surface structural flaws, identify ambiguous instances, and calibrate model evaluation pipelines. This aligns with findings from LM benchmarking literature, which suggest that compact, high-quality evaluation sets can offer meaningful diagnostic insights at significantly lower annotation cost. Nonetheless, the resource-intensiveness of expert involvement, especially in scientific and technical domains, remains a key constraint on dataset scalability and update frequency.

## K Uncertainty Quantification Details

**Overall Procedure.** For each original question we extract token probabilities for all options. The data is split into calibration and test sets from the calibration set and we compute conformal scores to determine a confidence threshold $\hat{q}$. On the test set, any option with probability exceeding $\hat{q}$ is selected as a prediction, allowing for a variable number of predicted options per question.

**Data Partitioning.** We partition the dataset into distinct calibration and test sets by randomly sampling asset types without overlap. The *calibration set* comprises: aero gas turbine, power transformer, pump, industrial gas turbine, and reciprocating internal combustion engine. The *test set* includes: fan, steam turbine, compressor, electric motor, and electric generator.

**Prompting Strategy.** We adopt the *Base Prompting* method, as described in prior work, where the input to the LLM consists of the complete question followed by all answer options. The model is instructed to return the correct choice, prefixed by "`Answer:`".

**Explaining Accuracy Variations.** Observed variations in accuracy for the same LLM across different setups (e.g., in comparison to PertEval) can be attributed to the following:

- Evaluation is conducted on a held-out test subset rather than the full dataset.

- Prompt format and phrasing differ from other benchmarks.
- In PertEval, answers are derived from structured JSON-based reasoning, while in our setup, predictions are based on direct option selection using model logits.

**Uncertainty-Adapted Accuracy (UAcc).** We define UAcc as:

$$\text{UAcc} = \frac{\text{Accuracy}}{\text{Set Size}} \sqrt{|\mathcal{Y}|}$$

where $|\mathcal{Y}|$ is the number of classes. In our reporting, UAcc is scaled by 100, and may exceed 100 in cases of high model certainty and small prediction sets.

**Confidence Level.** We set the error tolerance to $\alpha = 0.1$, ensuring that prediction sets include the true label with probability at least 0.9.

## L   Real World Applicability

### L.1   Electrical Transformer Predictive Maintenance

This dataset [27] consists of 48 timeseries variables collected from IoT devices placed in an electrical transformer from June 25th, 2019 to April 14th, 2020 which was updated every 15 minutes. We focused in predicting magnetic oil gauge faults.

### L.2   Air Compressor Predictive Maintenance

The Air Compressor dataset [23] consists of 19 sensor variables and 5 different failure modes: bearing, water pump, outlet valve, motor, and radiator failure indicators. For each failure mode, we provide the top-5 recommendations along with their correlations. We omit motor failure, because the dataset doesn't include any failure instance.

### L.3   Wind Mill Power Production Forecasting

The aim of this dataset [5] is to predict the wind power that could be generated from the windmill for the next 15 days across 20 sensor variables.

### L.4   Findings

For tasks like Air Compressor – Radiator failure prediction and Wind Mill – Energy Production forecasting, the top-ranked features exhibit strong correlations (up to 94.40%). These results demonstrate the LLM's ability to surface high-value variables for both forecasting and failure prediction. In contrast, for some failure tasks (e.g., Bearings, Valve), the LLM recommended low-correlation features which potentially reflects ambiguity or degraded signal quality. Interestingly, in one case (Magnetic Oil Gauge), the most correlated feature was ranked third, suggesting the LLM may prioritize semantic or contextual cues over raw correlation. These findings highlight both the promise and current limitations of LLM-driven feature selection in industrial settings.

**LLMFeatureSelect Input Prompt**

Select the variables from the list that are most relevant for predicting motor failure in air compressors. Provide the variables sorted starting with the one with the highest priority.

**Variables and their descriptions:**

**torque:** Torque is the turning force of a one-meter rod required to hold a 1kg mass constant

**outlet_pressure_bar:** The outlet pressure next to the piston valve

**air_flow:** amount of air that an air compressor can deliver

**noise_db:** Level of sound produced by an air compressor during operation

outlet_temp: temperature of the compressed air as it exits the compressor

**wpump_outlet_press:** water pump outlet pressure

**water_inlet_temp:** Water inlet temperature occurs according to the radiator size and fan capacity

**water_outlet_temp:** Water outlet temperature is related to how much of the heat generated by the heating of the compressed air is transferred to the water

**wpump_power:** Water pump power

water_flow: Water flow

**oilpump_power:** Oil pump power

**gaccx:** Ground acceleration in the x axis from where the compressor is attached to a rigid part or construction

**gaccy:** Ground acceleration in the y axis from where the compressor is attached to a rigid part or construction

**gaccz:** Ground acceleration in the z axis from where the compressor is attached to a rigid part or construction

**haccx:** Head acceleration in the x axis measured from the compressor head bolt or upper cooling fin. Each piece has a natural frequency and they oscillate at a certain frequency as a result of the incoming drive

**haccy:** Head acceleration in the y axis measured from the compressor head bolt or upper cooling fin. Each piece has a natural frequency and they oscillate at a certain frequency as a result of the incoming drive

**haccz:** Head acceleration in the z axis measured from the compressor head bolt or upper cooling fin. Each piece has a natural frequency and they oscillate at a certain frequency as a result of the incoming drive

**rpm:** Number of revolutions per minute of the electric motor

**motor_power:** amount of electrical energy that a motor can convert into mechanical power

**json:**
{"reasoning": "<your reasoning>", "selected_variables": ["variable 1", "variable 2", ..., "variable n"]}

---

**ChatGPT Response**

**Reasoning:**
Motor failure in air compressors is commonly associated with electrical overload, mechanical stress, excessive vibration, and thermal issues. The most indicative variables include motor power (to detect overloading or unusual consumption), torque (for mechanical resistance), and rpm (to observe speed fluctuations or stalling). Vibrations at the motor mounting (ground acceleration) and head (if driven directly) help detect mechanical imbalances, while noise can reflect internal wear or misalignment. A drop in air flow or changes in outlet pressure may signal downstream effects of motor underperformance.

**Selected Variables sorted by priority:**
motor_power, torque, rpm, gaccx, gaccy, gaccz, noise_db, haccx, haccy, haccz, air_flow

Figure 17: LLMFeatureSelect input prompt and ChatGPT response.

## M Human Evaluation via Expert Quiz

To assess the quality and realism of generated outputs, we conducted a human evaluation in the form of an expert quiz. The quiz was administered via Google Docs and designed to simulate real-world asset management decision-making tasks. Participants were domain experts who were provided with sufficient context, including the problem statement and the objective of the quiz.

Importantly, participants were not given access to any ISO standards or reference documents and were instructed to rely solely on their domain expertise. No external sources of information were permitted during the evaluation. Additionally, we did not disclose the source documents or datasets used to generate the quiz content, ensuring that judgments were based strictly on the information presented within the quiz itself.

The quizzes were not time-restricted, but we recorded the time each participant took to complete the 40 questions. Completion times ranged from 21 minutes to 90 minutes, with an average of 35 minutes, including questions where participants responded with "I don't know". The latter questions are not taken into consideration when calculating the participant's score.

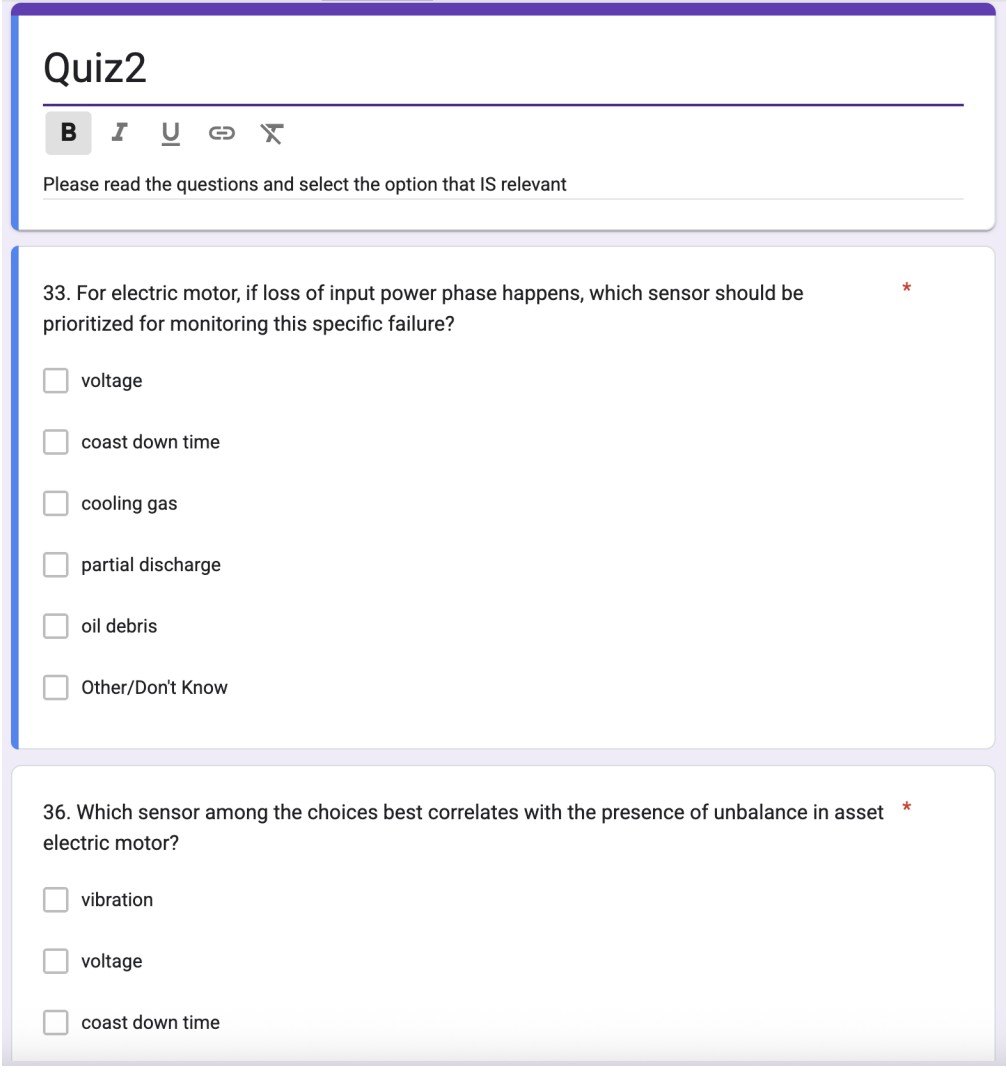

Figure 18: Screenshot from the quiz used for the user study.

