# OpenReview forum: "FailureSensorIQ: A Multi-Choice QA Dataset for Understanding Sensor Relationships and Failure Modes"
_NeurIPS.cc/2025/Datasets_and_Benchmarks_Track — NeurIPS 2025 Datasets and Benchmarks Track poster_

### Official Review · Reviewer_4Jsh · 2025-06-30

**Rating:** 4
**Confidence:** 3

**Summary:**

This paper introduces FailureSensorIQ, a domain-specific multi-choice QA benchmark designed to evaluate LLMs' reasoning ability in industrial diagnostic scenarios, especially around sensor-failure mode relationships. The authors target key Industry 4.0 challenges such as Condition-Based Maintenance (CBM) by leveraging structured knowledge from ISO documents and expert-annotated mappings.

**Dataset Code Accessibility:**

Yes

**Ethical Considerations:**

No, there are no or only very minor ethics concerns

**Limitations Weaknesses:**

1. The current dataset focuses only on static failure-sensor relationships. However, real-world predictive maintenance often involves temporal degradation patterns (e.g., time-series signals).

2. Although ReAct is used for tool-augmented inference, the retrieval reasoning loop is simplistic, and few insights are drawn about why performance does not improve.

3. The MC-MCQA (multi-correct) setting is described but results are offloaded to the appendix. Given its critical role in evaluating calibration and distractor handling, it deserves more attention in the main body.

4. The claim that asset-level performance correlates with online documentation is interesting but only weakly substantiated (a scatter plot in the appendix).

**Strengths Contributions:**

1. The benchmark targets a critically underexplored domain (industrial diagnostics) with a clear, real-world use case: mapping failure modes to sensor data. The SC-MCQA/MC-MCQA format derived from ISO documentation offers a rare, knowledge-rich testbed.

2. The Perturbation–Uncertainty–Complexity framework is a major strength. It goes beyond surface-level accuracy to quantify robustness, calibration, and reasoning limits under stress conditions. The observed fragility to distractors and performance variance under perturbations are critical insights for safe deployment of LLMs in industrial contexts.

3. Involving domain experts in the evaluation loop (with inter-rater agreement) is highly commendable. It provides strong grounding for comparing LLM performance against human baselines. The ReAct agent evaluation uncovers a misalignment between external retrieval and performance gains, emphasizing the complexity of tool-use in domain-specific QA.

---

> ### Author Rebuttal · Authors · 2025-07-31
>
> We thank the reviewer for their valuable feedback.
>
> ----
>
> **Comment: 1 Incorporating the temporal signature from Sensor**
>
> **Response:**
>
>
> Given that LLMs are still in the early stages of understanding time series primitives—such as "high", "low", "increasing", and "decreasing"—in conjunction with domain-specific signals, we have identified this as a future direction in our literature review. We are committed to scientific innovation, and the following outlines our planned course of action.
>
> In Section 4, we discussed a ReAct-based experiment. As part of this experiment, we tracked which tools were invoked, which queries were issued, and what passages were retrieved. Nearly 15,000 valuable passages were collected, and we hope these passages will provide the kind of temporal understanding required to generate appropriate QA pairs. After verifying their usefulness and correctness with our SMEs, we may decide to gather real or simulated time series data. We have included one example for reference, where reader can see clearly the complexity involved in understanding and interpreting the signal.
>
> Example from fetched paper from Arxiv:
>
> Paper: Physical Variable Measurement Techniques for Fault Detection in Electric Motors
>
> Abstract: ... This work finds that current and vibration are the most popular signals employed to detect faults in induction motors. However, stray flux signal analysis is presented as a promising alternative to detect faults under certain operating conditions where other methods, such as current analysis, may fail.
>
>
> ----
>
>
>
> **Comment: 2 Although ReAct is used for tool-augmented inference, the retrieval reasoning loop is simplistic, and few insights are drawn about why performance does not improve.**
>
> **Response:**
>
>
> We present two pieces of evidence to support our claim. The first is based on our own analysis of ReAct trajectories (think-act-observe cycles), and the second is drawn from independent research observations in the community.
>
> **Observation 1**
>
> Our ReAct agent is a robust implementation that integrates two external tools from the LangChain repository—Wiki and ArXiv—alongside an internal Self-Ask module that supports chain-of-thought reasoning. Using the standard ReAct prompt from LangChain's Prompt Hub, extended with Self-Ask, the agent performs both external searches and internal reasoning. This implementation can be found by searching Google for “ReActXen IBM”.
>
> We generated nearly 13,000 ReAct trajectories (2667 queries × 5 options) to produce the results reported in Figure 7. To better understand tool effectiveness, we developed a trajectory investigation framework that includes a credit assignment system, which attributes credit to the tools most responsible for generating the final answer.
>
> The table below summarizes the average tool credit (± standard deviation) for each model across Wiki, ArXiv, and Self-Ask tools. One clear pattern emerges: external tools consistently received lower credit than the internal Self-Ask reasoning module, indicating a gap in when to search, what to search and reason over search result. Self-Ask means LLM’s internal knowledge is mostly used to derive the answer.
>
> **ReAct Agent Tool Credit Assignment Summary**
>
> Each row represents a different Llama-based model. The columns show the average ± standard deviation of tool credit for Wiki, Arxiv, and Self-Ask tools across 2667 ReAct trajectories.
>
>
>
> | Model               | Wiki (avg ± std)     | Arxiv (avg ± std)    | Self-Ask (avg ± std) |
> |---------------------|----------------------|-----------------------|-----------------------|
> | Llama-3.1-8B-Instr          | 55.25 ± 17.65        | 37.55 ± 15.78         | **76.84± 10.95**         |
> | Llama-3.3–70B-Instr         | 11.60 ± 8.59        | 8.52 ± 5.07         | **80.97 ± 7.46**        |
> | Llama-3.1-405B-Instr           |  69.97 ± 12.23        | 28.11 ± 8.70         | **77.10 ± 2.98**       |
> | Llama-Maverick            | 22.04 ± 13.93        |  16.49 ± 8.41         | **64.22 ± 14.33**       |
> | Llama-Scout | 22.51 ± 14.72       | 16.38 ± 8.37         | **64.55 ± 14.17**        |
>
>
>
> **Observation 2 (From community supporting our conclusion)**
>
> In the paper Page 8, in the last three lines we present our findings – 'method for searching and reasoning is critical – and currently underexplored', We have similar scientific observations motivated in an article made available on arxiv on 25 June 2025 [1]. This shows an additional support that there is a gap in LLMs and which is what we experienced.
>
> [1] MMSearch-R1: Incentivizing LMMs to Search (See Page 2 – "….highlight three aspects, when to search, what to search and reason over search result…") – 25 June 2025
>
>
> ----
>
>
>
> **Comment: 3 The MC-MCQA (multi-correct) setting is described but results are offloaded to the appendix. Given its critical role in evaluating calibration and distractor handling, it deserves more attention in the main body.**
>
> **Response:**
>
>
> We agree with the reviewer on the importance of the MC-MCQA dataset discovery and will make adjustment into the main paper swapping out figure 3. We also have more result responsed. However, we would like to share our rationale for initially placing it in the Appendix:
>
> Ease of reading and clarity – The evaluation metrics for SC-MCQA and MC-MCQA differ significantly. To avoid potential confusion, we chose to keep the MC-MCQA results separate from the main evaluation.
>
> Complementary evidence – The results from the Uncertainty and Complexity analysis (Table A) indirectly support our observations from the MC-MCQA evaluation (Table 15). As question complexity increases—either by expanding the option pool (e.g., from 5 to 10) or by requiring multiple correct answers—the model’s performance declines, highlighting the increased difficulty.
>
> Metric inconsistency – Our uncertainty-based method allows the model to select a variable number of options until a confidence threshold is met. This introduces a different evaluation metric compared to fixed-choice setups (see Section 3.2 vs. Appendix Table 15). To maintain consistency in evaluation frameworks, we placed this content in the appendix.
>
>
> ----
>
>
>
> **Comment 4: The claim that asset-level performance correlates with online documentation is interesting but only weakly substantiated (a scatter plot in the appendix).**
>
> **Response:**
>
>
> We would like to clarify that the scatter plot analysis is not only presented in the Appendix—it is also included in Figure 6 of the main text.
>
> A challenge that we face is that the model’s training data is typically not publicly available. As a result, it is difficult to know what information the model has already seen.
>
> Furthermore, many of the knowledge sources indexed by the model contain a vast amount of information, and our current retrieval strategy relies on keyword-based search. This can be limiting when trying to precisely locate relevant technical content.
>
> We are actively working to expand the asset class coverage from 10 to 53. Once this expansion is complete, we will be in a better position to quantify coverage more rigorously and provide a statistically meaningful analysis.
>
> Our motivation for incorporating ReAct, especially using arXiv as a source, was largely driven by the analysis shown in Figure 6. However, conducting searches over such a large and diverse information space has proven to be a challenging problem in its own.

---

> > ### Comment · Reviewer_4Jsh · 2025-08-05
> >
> > Thank you for the clarifications. I acknowledge the new results and appreciate the response. I will keep my score as is and look forward to seeing the improvements reflected in the camera-ready version.

---

> > > ### Author Response · Authors · 2025-08-05
> > >
> > > Thank you very much for your time and acknowledging our rebuttal. We will revise the paper with all the additional result we included during rebuttal and discussion period.

---

### Official Review · Reviewer_1uGp · 2025-07-02

**Rating:** 5
**Confidence:** 2

**Summary:**

This paper studies the problem of evaluating the capability of large language models (LLMs)to understand sensor relationships and failure modes in industrial systems, and introduces a benchmark dataset comprising 8,296 multiple-choice questions across 10 industrial assets, generated from ISO documents and curated using expert-designed templates. Several LLMs are evaluated on this dataset under various conditions, including perturbation, complexity, and RAG settings, to assess their reasoning robustness and domain-specific knowledge.

**Dataset Code Accessibility:**

Yes

**Ethical Considerations:**

No, there are no or only very minor ethics concerns

**Final Justification:**

In general, this paper addresses an interesting problem and presents sufficient experiments to support its exploration. My earlier concerns regarding the limited problem scope (restricted to MCQA) and the unclear description of negative option generation have been addressed during the rebuttal. Overall, I remain positive about the acceptance of this paper.

**Limitations Weaknesses:**

I find this paper largely complete and only have some minor problems:

- The current question format in the dataset is limited to selecting relevant sensors for given failure modes or vice versa. It would be great to include more diverse problem types, such as true/false questions, open-ended questions, or ranking tasks, to better evaluate the breadth of LLM capabilities in industrial diagnostics.
- The method used to generate negative options is not clearly described. If distractors are sampled randomly, it may result in overly simplistic or uninformative questions.

**Strengths Contributions:**

- The studied problem is interesting and critical in the downstream industrial applications of LLMs. The current results presented in this paper reveal the limitations of existing LLMs, as they still fail to correlate the correct sensors to different failure modes.
- Overall, the experiments are thorough and encompass a wide range of settings to effectively evaluate LLMs under diverse conditions.

---

> ### Author Rebuttal · Authors · 2025-07-31
>
> We thank the reviewer for their contribution, support, and valuable questions.
>
> ----
>
>
> **Comment: 1 The current question format in the dataset is limited to selecting relevant sensors for given failure modes or vice versa. It would be great to include more diverse problem types, such as true/false questions, open-ended questions, or ranking tasks, to better evaluate the breadth of LLM capabilities in industrial diagnostics.**
>
> **Response:**
>
> We conducted two set of benchmarks a) true/false questions and b) open-ended questions.
>
> **True/False question created and performance study:**
>
> We prepared a True/False version similar to TruthfulQA [1], comprising 2,995 questions (to be made available). For each question, we provide a statement describing the relationship between an asset’s failure mode and a sensor and then ask whether it is true or false. For each asset and failure mode and sensors combination we construct a question. We run this dataset through the Perturbation [2] and Uncertainty [3] evaluation pipelines, as described in Section 3. We report results for several 8B models. Answers with invalid formatting are considered incorrect.
>
> **Perturbation Analysis**: (Similar to Figure 4 in paper)
>
> |Model | ACC@Original | ACC@Simple | ACC@Complex | Consistency |
> |-------------------------|--------------|------------|-------------|-------------|
> | Llama-3.1-8B-Instr | 49.30 | 47.07 | 42.52 | 27.38 |
> | Qwen3-8B | 65.71 | 64.90 | 63.88 | 49.26 |
> | Mistral-8B-Instr-2410 | 49.63 | 48.78 | 44.21 | 26.26 |
> | Granite-3.3-instr | 58.81 | 34.57 | 52.64 | 21.52 |
>
> **Uncertainty Quantification**: (Similar to Table 3 in paper)
>
> |Model| UAcc| SS | CR | Acc |
> |------------------------|-------|------|-------|--------|
> | Llama-3.1-8B-Instr | 88.06 | 1.82 | 95.11 | 65.14 |
> | Qwen3-8B | 47.44 | 1.89 | 93.03 | 36.57 |
> | Mistral-8B-Instr-2410 | 88.53 | 1.79 | 95.17 | 63.91 |
> | Granite-3.3-Instr | 81.75 | 1.80 | 94.56 | 59.57 |
>
> Accuracy is naturally higher since there are only 2 options when compared to the original multiple-choice which mostly has 5 options. We observe a similar phenomenon of performance degradation with simple and complex perturbation of the prompt. Overall, performance on original > perturb simple > perturb complex, with only exception of granite which had severe performance degradation on perturbed complex questions.
>
> Overall, the consistency is below 50%, and the Set Score Size (SS) is close to 2, despite the Uncertainty Accuracy (UAcc) being higher than 80%. This indicates that in the majority of cases, the model is highly uncertain between the two options. These results will be included in the Appendix, along with a mention of additional datasets as part of our key contributions. We encountered some difficulty with the Qwen3-8B model during the uncertainty evaluation. This is due to the `<think>` output format, and the fact that the underlying implementation does not yet support reasoning-based models.
>
> **Open Ended Question Creation and its performance study.**
>
> **Generation**. We manually created 88 open-ended questions (to be made available), such as: “List all failure modes of an electric motor that can be detected by **vibration**, **cooling gas**, or **axial flux** sensors.” These open-ended questions can vary in complexity, especially when combining multiple aspects such as different sensor types, as shown in the example or assets or failure modes. The answer to all the questions is in a list form and the average number of items in the ground truth set is **5.7**.
>
> **Evaluation**. Open ended question is complex problem than MCQA from generation and evaluation purpose. Such questions may yield a single answer or responses long enough to fill the entire context window. To address this, our system prompt (to be included as part of the paper) explicitly provides guidance: “Include only 1 to {item_count} items in a Python-style list, e.g., ["answer 1", "answer 2"],” where {item_count} is dynamically set to **5** or **10** based on the evaluation setting. We adopt a two-phase approach: a generation phase to generate potential candidate answers, followed by an extraction phase, which structures the output into a valid JSON format. We then compare the generated list with the ground truth using structured semantic entity evaluation metrics for evaluation [4] and obtained precision and recall.
>
> To assess the generation performance, we experimented with five models—including the Llama family, our traditional method, and the Mistral family—on this new set of open-ended questions.
>
> **Model Performance: Precision@5 means that item_count was set to 5 in generation prompt.**
>
> | Model | Precision@5 | Recall@5 | Precision@10 | Recall@10 |
> |-|-|-|-|-|
> | Ministral-8B-Instruct | 0.2877      | 0.0563   | 0.2144        | 0.0695    |
> | mistral-medium-2505   | 0.3273      | 0.2545   | 0.1778        | 0.2305    |
> | **mistral-large**     | **0.3826**  | **0.4089** | 0.2349      | **0.3945** |
> | Llama-3.1-8B          | 0.1955      | 0.0367   | 0.1216        | 0.1018    |
> | llama-3-3-70b         | 0.3205      | 0.1624   | **0.2812**    | 0.2022    |
> | llama-3-405b          | 0.2554      | 0.2407   | 0.1933        | 0.2739    |
> | llama-4-maverick      | 0.3251      | 0.2808   | 0.2579        | 0.2826    |
> | granite-3-3-8b        | 0.0732      | 0.0283   | 0.1086        | 0.0486    |
> | Qwen-3-8b             | 0.3229      | 0.1301   | 0.2292        | 0.0874    |
>
> Increasing the number of candidates from top-5 to top-10 did not significantly improve recall and consistently reduced precision. This indicates that most relevant results are already captured within the top-5, and additional candidates introduce noise. We noticed the mistral family’s generation ability is better than llama and Qwen-3-8b a reasoning model is also performing well. We will include an additional section in Appendix to capture this work-thread.
>
> [1]  TruthfulQA: Measuring How Models Mimic Human Falsehoods, ACL 2022
>
> [2] PertEval: Unveiling Real Knowledge Capacity of LLMs with Knowledge-Invariant Perturbations, NeurIPS 2024 D&B
>
> [3] Benchmarking LLMs via Uncertainty Quantification, NeurIPS 2024
>
> [4] Structured Document Generation for Industrial Equipment, AAAI-2024
>
>
>
> **Ranking task**
>
> Ranking the sensors’ priority is an interesting future direction but also has a challenge. Failure modes have different root causes that can vary the sensors’ priority and we can work with our SMEs in the future to create such a task.
>
>
> ----
>
>
>
> **Comment: 2 The method used to generate negative options is not clearly described. If distractors are sampled randomly, it may result in overly simplistic or uninformative questions.**
>
> **Response:**
>
>
> A due diligence was taken on selecting the distractors, it was not purely random. We used our Reliability Strategy Library to gain failure modes for different assets. Our experts identified plausible sensors for monitoring each asset (they vary per asset) and labeled the failure modes and sensors that are related for that asset. Then for each (asset, failure mode, sensor) triplet they marked it as positive if for the given asset the failure mode and sensors are related, otherwise as negative. This is the table $D_{relevant}$ that we mention in Appendix B. We provide an example in Table 1 for a given asset.
>
> To construct a single-correct multiple choice, for each asset, failure mode and positive sensor, we sample negative sensors that are plausible for that asset and use the question template to form a question. We repeat the same procedure for failure modes. We follow the same procedure for multi-correct multiple choice but use pairs of related items instead. The code for constructing this multiple-choice is available in our supplementary materials and on GitHub. We will be updating Appendix B (Section B.2, Figure 8, Algorithm 1) to describe this more in detail.

---

> > ### Comment · Reviewer_1uGp · 2025-08-04
> >
> > I am satisfied with the responses and have no further questions.

---

> > > ### Author Response · Authors · 2025-08-04
> > >
> > > We sincerely thank you for your valuable review and for acknowledging our rebuttal.

---

### Official Review · Reviewer_UAK9 · 2025-07-05

**Ethics Flags:** Safety and security
**Rating:** 4
**Confidence:** 4

**Summary:**

This paper proposes FailureSensorIQ, a multiple choice question and answer (MCQA) benchmark dataset and evaluation framework for industrial asset troubleshooting. FailureSensorIQ is aiming to evaluate the capabilities of large language models (LLMs) for reasoning about sensor-failure mode relationships. Starting from ISO documents and expert knowledge, the authors automatically generate a total of 8,296 questions (in both single-correct and double-correct formats) for 10 assets, and systematically reveal current LLM's performance upper limits and weaknesses in this domain through multi-dimensional experiments (perturbation robustness, Uncertainty, Complexity, etc.). The paper also presents LLMFeatureSelector, an LLM-based feature selection scikit-learn pipeline tool, and validates it on three real fault prediction datasets.

**Dataset Code Accessibility:**

Yes

**Dataset Code Comments:**

The Dataset Reviewer Report looks good, the missing license is added. The paper is targeting the problem of lacking the multi-choice Q&A datasets for sensor-failure mode inference for industrial assets, an important gap for the industrial asset troubleshooting. You propose a FailureSensorIQ, a multiple choice question and answer (MCQA) benchmark dataset and evaluation framework for industrial asset troubleshooting. Especially, we request further clarification and additional experimental evidence:

Asset Selection and question scope Have you quantified question difficulty per asset (e.g., complexity scores, expert response times)?

Did you ensure a balanced difficulty distribution across different assets to enable fair model comparisons?

You selected ten industrial assets. How were these chosen to represent the broader industrial domain? If one were to include other asset types, can your generation pipeline and templates seamlessly adapt? Please provide qualitative or quantitative evidence of such extensibility.

Multi Answer MCQA Evaluation Details For MCQA, your EM rate is below 21%. How many answers are models instructed to select? Do you explicitly inform the model of the number of correct choices, and how does this affect performance? Please compare scenarios where the model knows and does not know the answer count.

Real-World Applicability Your dataset is entirely derived from ISO standards without any real sensor logs or failure records. How do you ensure that models trained on FailureSensorIQ will generalize to noisy, heterogeneous field data? Can you provide pilot experiments or case studies using real-world sensor traces to validate transferability?

Industrial assets often exhibit domain shifts—different manufacturers, operating conditions, and sensor calibrations. Have you assessed how asset specific idiosyncrasies (e.g., sensor scale differences, fault mode prevalence) affect model performance? Please discuss any robustness tests or domain‑adaptation strategies you’ve considered.

Limited Expert Evaluation Sample In Section 3, you report human validation by only five evaluators on 80 questions (under 1% of the 8,296 item set), which may not adequately capture inter-annotator agreement or task reproducibility across your ten asset types. Could you justify how this limited sample size still

ensures dataset quality? Please provide additional agreement statistics on a larger or more diverse subset.

Could you describe the criteria for selecting your expert reviewers to demonstrate the robustness and repeatability of your validation process?

**Ethical Considerations:**

No, there are no or only very minor ethics concerns

**Limitations Weaknesses:**

-	Responsible AI metadata is lacking: The lack of RAI information on data collection, bias, labeler background, and social impact does not allow for a review of data fairness and potential risks, and makes it difficult to guide compliance considerations in actual deployments.The writing needs to be improved.
-	Limited authenticity of “semi-synthetic” data: Q&A pairs are automatically generated based on predefined templates and expert mapping, which ensures the openness and reproducibility of the data. However, this means that it may not fully reflect the noise distribution and anomaly patterns of sensor readings in real production environments.
-	Lack of Time Series and Dynamic Reviews: As the paper describe, the current version only examines static questions and answers, and does not include dynamic reasoning about sensor-fault evolution over time.

**Strengths Contributions:**

-	Novel Focus: The first presentation of a multi-choice Q&A dataset for sensor-failure mode inference for industrial assets, distinguishing it from both generalized QA and pure time-series prediction, and providing a new benchmark for LLM in the direction of industrial inference.
-	Comprehensive and meaningful experiments: Experiment include multi-dimensional examination from model size, cueing strategy, robustness, complexity and supplemented with human expert and ReAct Agent. So the conclusions are persuasive.
-	Open and Reproducible: The dataset, loading scripts, and evaluation code have been published on GitHub, making it easy for the community to quickly reproduce, extend, and compare other methods.

---

> ### Author Rebuttal · Authors · 2025-07-31
>
> We thank the reviewer for their thorough review and their insightful questions.
>
> -----
>
> **Comment: 1 Responsible AI metadata is lacking**
>
> We leveraged our Reliability Strategy Library (RSL) which has information for hundreds of assets and thousands of failure modes, to get the list of failure modes, and then our experts with a diverse background in mechanical and electrical engineering provided us the Failure Mode and Sensors mapping.
>
> As noted on **Page 9, Line 1**, the first paragraph highlights the role of expert input in the validation process, with additional details provided in **Appendix Section G**.
>
> The composition of our industrial product team is:
> - Asset Performance Management (APM) – Focuses building solutions for physical asset lifecycle management, composed of **reliability engineers** and **professional engineers**.
> - Operational Site Management – **Site managers** and **plant operators** with **years of hands-on experience** managing critical infrastructure in **data center environments**.
> - Industrial AI Community – **Industrial data scientists and SMEs** involved in deploying AI solutions in domains such as **Oil & Gas**.
>
> We will include in Section G the **diversity of expertise**, particularly across **4 geographical regions** and **gender representation**.
>
> We reviewed past year's DB-track papers and followed their conventions.  Section G provides essential details about our process aligned with prior art—such as the **number of participants** and their areas of expertise—which directly informed several experimental design decisions, including the selection of evaluation questions and scenario configurations.
>
> Social impact: increase productivity of engineers by reducing asset failures, reduce money spending on maintenance, and increase the reliability in a variety of business types.
>
> **Comment: 2 Limited authenticity of “semi-synthetic” ... real production environments.**
>
> Currently, we are focused on automating the identification of the right set of monitoring parameters for each failure mode.
>
> Before we embark on leveraging real-time streaming data, it is important to assess **how much foundational knowledge an LLM possesses about an asset** from a **common-sense or first-principles** perspective—like baseline evaluations in a medical study (Ref. 18, Can LLM Reason about Medical Question). This step serves as a critical foundation for downstream diagnostic reasoning.
>
> **Comment: 3 Lack of Time Series and Dynamic Reviews**
>
> Our decision to focus on this task was driven primarily by the availability of data suitable for constructing a comprehensive benchmark.
>
> Once time-series data is available, it enables automated anomaly detection workflows—including **model selection** and **parameter tuning**—which are essential for addressing **false alerts**, one of the most persistent challenges in monitoring systems.
>
> While traditional time-series models are adept at capturing temporal patterns, they often fall short in contextual reasoning. In contrast, **Large Language Models (LLMs)** excel at **semantic interpretation** and can infer **structured domain knowledge**, such as the relationships among failure modes, assets, and sensors.
>
> **Comment: 4 License**
>
> There is a LICENSE file present in main repository that we have provided at the time of submission.
>
> **Comment: 5 Asset Selection and question scope**
>
> Due to space constraints, we would kindly refer you to Reviewer ```R1pw``` response on "reasons behind choosing 10 industrial assets".
>
>
> **Comment: 6 Question Scope**
>
> We focus in Failure Mode to Sensors and Sensors to Failure Modes questions because they have important applications in industrial asset monitoring. As stated in the **Introduction**: A key application within Industry 4.0 is Condition-Based Maintenance (CBM), which focuses on monitoring the health of assets using sensor data.
>
> **Comment: 7 Question difficulty quantification**
>
> We will propose a complexity measure; however complexity in our domain doesn’t necessarily correlate with the LLMs performance due to memorization.
> Complexity measure composes of:
> - Asset Complexity (C): We can rank our assets based on a complexity scale from 1 to 3 based on number of components and interactions.
> - Asset’s a Failure Mode (FM) / Sensor (S) out degree deg(FM_ai)/deg(S_ai): If it’s very low, then there is less confusion/ambiguity for the answer giving it a lower complexity score. Normalized by asset.
>
> For failure mode question i of asset a the overall formula would then be: α * C + β * deg(FM_ai) where α and β are weighting factors depending on where to give more emphasis for complexity. Finally, this measure can be normalized across a scale (e.g., out of 5).
>
> **Question response times:**
>
> The quizzes were not time-restricted, but we recorded the time each participant took to complete the 40 questions. Completion times ranged from 21 minutes to 90 minutes, with an average of 35 minutes, including questions where participants responded with "I don't know."
>
> **Comment: 8 How representative are the 10 assets? Is your pipeline extensible?**
>
> We cover 10 more general assets that are core for the operation of many critical businesses.
>
> We have SMEs which helped in the development and correctness verification of this dataset on these assets with decades of experience on these asset classes. The procedure can be generalizable into any asset classes, as long as we get access to SMEs of that particular asset class.
>
> We have 9000 additional questions from another 35 asset classes that will be manually verified by SMEs to extend our dataset in the future. Some additional assets we plan to include are Switchgear, Pressure vessels and Valves.
>
> **Comment: 9 Multi Answer MCQA Evaluation ... compare scenarios where the model knows and does not know the answer count.**
>
> We did not explicitly inform the models of the number of correct choices to align with real-world scenarios, where such information is not provided which is harder hence low exact match (EM) rates. We conducted an additional experiment in which we explicitly mention the number of correct options where we observe that LLMs perform better.
>
> | Model | EM | F1 | Set Size |
> |-|-|-|-|
> | o3 | 0.384 | 0.635 | 2.0 |
> | o4-mini | 0.382 | 0.633 | 2.0 |
> | gpt-4.1 | 0.385 | 0.633 | 2.0 |
> | gpt-4.1-mini | 0.385 | 0.635 | 2.0 |
> | gpt-4.1-nano | 0.384 | 0.634 | 2.0 |
> | llama-4-maverick | 0.434 | 0.678 | 2.0 |
> | Llama-4-scout | 0.403 | 0.661 | 1.99 |
> | llama-3-1-405b | 0.376 | 0.635 | 2.0 |
> | llama-3-3-70b | 0.343 | 0.610 | 2.0 |
> | Llama-3.1-8B | 0.252 | 0.561 | 2.16 |
>
>
> **Comment: 10 How do you ensure ... transferability?**
>
> One point we would like to clarify is that our work is solely focused on introducing the FailureSensorIQ dataset and benchmark; we do not train or fine-tune LLMs in this paper. It is out of scope for this paper and we have a separate line of our research work.
>
> To assess the transferability, we are actively conducting pilot case studies using proprietary datasets that span 27 different asset types with 400 real sensor traces. We evaluate multiple LLMs using both open-ended and multiple-choice QA formats. We observe that model performance on these real-world datasets closely mirrors the rankings on the FailureSensorIQ benchmark. This provides early empirical evidence that FailureSensorIQ effectively captures critical reasoning patterns in real-world scenarios.
>
> Also, we have reported in paper the similar work. **See Table 8** - where we apply our idea on 5 Asset class and reported interesting results.
>
> **Comment: 11 “Real-World Applicability Your dataset is entirely derived from ISO standards without any real sensor logs or failure records.”**
>
> In this work we followed a more theoretical approach where we used candidate failure modes and sensors for each asset from our RSL. We then consulted SMEs to verify.
> We avoided timeseries analysis to derive these relationships because of several real-world challenges:
> - Severely missing and noisy data often found in the real world that would make it harder for us to make any conclusions.
> - Failure modes have varying frequency, with some being very rare (once in a lifetime of an asset which can be several years only IF it occurs).
> - Failure dates and reports need to be available to trace patterns.
>
> **Comment: 12 Industrial assets often exhibit domain shifts—different ... robustness or domain‑adaptation strategies you’ve considered.**
>
> We do agree with this observation. An asset will exhibit different behavior under different context.
> We start with the more general case and build a repeatable pipeline and then extend it with thousands of failure modes for hundreds of different assets under different operating conditions.
>
> **Comment: 13 Limited Expert Evaluation ... a larger or more diverse subset.**
>
> Our 80 sample is drawn from Single Correct MCQA which comprises of 2667 (**around 3%**). As described in Section 4 in user study details, we draw a balanced sample across assets, relevancy type (selection/elimination), question type (failure mode to sensor / sensor to failure mode). This is an intensive quiz where every option requires thinking, and we didn’t want to overwhelm our SMEs with more questions, which could impact their performance to the quiz. In **section G** we provide some more details about the experts and discuss what other popular benchmark papers did for human evaluation.
>
>
>
> **Comment: 14 Could you describe the criteria for selecting your expert reviewers to demonstrate the robustness and repeatability of your validation process?**
>
> As mentioned in section G, the evaluators are experts in IBM’s industrial product team (reliability engineer with Band 7) and have helped in assessing the dataset’s correctness and difficulty and have an extensive knowledge about these assets’ operations.
>
> We have outlined the team composition in the Responsible AI question above.

---

> > ### Comment · Reviewer_UAK9 · 2025-08-05
> > **Thanks for the reply**
> >
> > Thanks for the reply, especially for the new results on the justification for the expert evaluation sample size, the details on how the sample was balanced, the supplementary evaluation of MC-MCQA  and the difficulty classification metrics for questions. Please consider adding these results to the paper. Regarding the problem of the dataset's semi-synthetic nature and its generalizability to real-world, noisy data, I am not yet convinced by your response. If you do not directly present a methodology or preliminary evidence for your plan to extend the pipeline to noisy, real-world data, then it should not be mentioned as a justification for the current paper's limitations.
> >
> > I am keeping my score, as I believe it fairly reflects the contribution of the paper.

---

> > ### Author Response · Authors · 2025-08-05
> >
> > Thank you for acknowledging that our rebuttal was able to address the majority of the points that you raised. We are going to add all the results reported as part of the rebuttal. Due to space limitations, some of the sections that you highlighted were trimmed. Now we have more space to give you more details on our plans.
> >
> > In Section 4, we discussed a **ReAct-based** experiment. As part of this experiment, we tracked which tools were invoked, which queries were issued, and what passages were retrieved. Nearly 15,000 valuable passages (written by scientists and etc) were collected. We hope these passages will provide the kind of temporal understanding required to generate appropriate QA pairs. After verifying their usefulness and correctness with our SMEs, we will try to gather sensor time series data that reflect the described behavior. We have included one example for reference, where the reader can clearly see the complexity involved in understanding and interpreting the signal. We did discuss this point in response to reviewer ``4Jsh``. Please take a deeper look.
> >
> > Example from a fetched paper from Arxiv:
> >
> > ```
> > **Paper**: Physical Variable Measurement Techniques for Fault Detection in Electric Motors
> > **Abstract**: ... This work finds that current and vibration are the most popular signals employed to detect faults in induction motors. However, stray flux signal analysis is presented as a promising alternative to detect faults under certain operating conditions where other methods, such as current analysis, may fail.
> > ```
> >
> > This passage is grounded and written by an expert, a key requirement for conducting any testing. We will be updating limitations and future work section to clarify this.
> >
> > We hope this has answered your question, please do not hesitate to follow up if necessary.

---

### Official Review · Reviewer_R1pw · 2025-07-06

**Rating:** 5
**Confidence:** 3

**Summary:**

The paper introduces FailureSensorIQ, a new multi-choice question-answering dataset (8,296 questions) designed to evaluate LLMs on their understanding of sensor relationships and failure modes across 10 industrial assets for Industry 4.0. It includes two diagnostic mapping tasks: Failure Modes to Sensor Relevance (FM2Sensor) and Sensor to Failure Mode Relevance (Sensor2FM). Experimental results show that various LLMs are fragile to perturbations, distractions, and uneven knowledge availability. This indicates the need for reasoning-aware and agentic LLM systems tailored to industrial diagnostics.

**Additional Feedback:**

- I am curious why the authors claim FailureSensorIQ is a benchmarking "system," not just a dataset.

**Dataset Code Accessibility:**

Yes

**Dataset Code Comments:**

The GitHub repo contains detailed documentation and readily usable data.

**Ethical Considerations:**

No, there are no or only very minor ethics concerns

**Final Justification:**

The authors have adequately addressed my concerns.

**Limitations Weaknesses:**

- What is the very motivation to have a unified LLM benchmark to solve these tasks if these tasks have been already tackled in the real-world applications without LLMs?
- What are the reasons behind choosing 10 industrial assets?
- What are some asset-specific challenges?

**Strengths Contributions:**

- The paper is clearly written and easy to understand.
- FailureSensorIQ addresses a critical gap between Industry 4.0 and recent AI advances by introducing a specialized benchmark for LLMs in predictive maintenance and sensor fault detection.
- The introduction of FM2Sensor and Sensor2FM tasks is novel.
- The analysis is rigorously discussed involving various LLMs across various settings.

---

> ### Author Rebuttal · Authors · 2025-07-31
>
> We sincerely thank you for your time and valuable feedback. In this rebuttal, we take the opportunity to further clarify the impact of our work on advancing autonomous agents for Industrial AI.
>
> -----
>
> **Comment: 1** Motivation to have a unified LLM benchmark to solve these tasks if these tasks have been already tackled in the real-world applications without LLMs?
>
> **Response**:
>
> In the abstract, we made a bold proposition: complementing **traditional data-driven analysis** with a **knowledge-driven approach**, where knowledge may originate from **human experts** or **LLMs** with **reasoning behind the decisions**. This framing acknowledges that, especially in industrial domains, expert-derived insights and structured reasoning play a vital role beyond what raw data can reveal. We further elaborate this transition in **Section 1.1**, where we outline the limitations of conventional data-centric pipelines and describe a more integrated view of domain knowledge as a first-class citizen. Additionally, **Section 4** demonstrates a small-scale application built using this principle—showcasing how LLMs and expert rules can complement sensor data to improve interpretability and decision-making.
>
> One of the core applications of Industry 4.0 has been the development of predictive and condition-based maintenance systems using real-time data streams. The author of this papers have significant hands-on experience building predictive models for industrial assets such as chillers, wind turbines, oil wells, compressors and more. In these projects, essential inputs—such as feature selection and failure mode understanding—have traditionally come from subject matter experts (SMEs).
>
> However, relying on SMEs is often not scalable. Their availability is limited—especially during time-sensitive engagements—and extracting domain knowledge from them repeatedly is not sustainable. One remarkable response from industry has been to codify field expertise into structured libraries such as the Reliability Strategy Library (RSL)/Failure Mode Effect Analysis(FMEA) tool. The IBM's RSL covers over **800** asset classes and includes more than **54,000** failure modes, providing a comprehensive mapping from asset to failure modes.
>
> Yet, a critical gap remains: which sensors or parameters help detect each failure mode, and how to use them for root cause analysis. Public datasets like those on Kaggle typically cover just one failure mode per asset, occasionally extending to 5–10, which is far from sufficient to build up such mapping knowledge base (See supporting evidence towards the end of review). In practice, mapping failure modes to detection logic has largely depended on discussions among a small group of domain experts.
>
> **Scaling Industrial Knowledge: From FMEA to Autonomous Agents**
>
> A critical challenge lies in extending traditional **FMEA** (Failure Modes and Effects Analysis) frameworks by integrating these mappings to support **Condition-Based Maintenance (CBM)** and **Root Cause Analysis (RCA)**. Our dataset is built with this vision in mind: aligning failure mode libraries with **AI-native tools** to accelerate predictive maintenance workflows across a diverse set of asset classes.
> In summary, our work brings a **unique value proposition**—not only to the field of reliability engineering but also to the broader
>
> **AI research community**. Just as AI has transformed the medical domain, we argue that **Industry 4.0 deserves similar academic attention**, especially given its potential to **enhance safety** in hazardous environments and enable **autonomous agents for critical industrial tasks**.
>
> **Supporting evidence:**
>
> We analyzed **135 publicly available industrial awsome datasets**—many sourced from platforms like Kaggle and the UCI Machine Learning Repository. The motivation was to evaluate them through the lens of **failure mode coverage**, as described in their dataset documentation or challenge descriptions.
>
> Despite the diversity of assets and use cases, a common pattern emerges: most datasets focus on **single failure mode analysis**, limiting their applicability in real-world, large-scale predictive maintenance systems. This highlights a **scalability gap in the literature** when it comes to handling multiple concurrent failure modes per asset.
>
> **Summary Statistics:**
> - **Total datasets analyzed**: 135
> - **Datasets with work order information**: 1
> - **Datasets with time series sensor data**: 84
> - **Datasets explicitly mentioning failure modes**: 53
> - **Datasets with alert/alarm signals**: 3
> - **Failure Mode Count Distribution:**
>      ```{0: 93, 1: 18, 2: 9, 3: 4, 4: 5, 5: 2, 6: 1, 7: 1, 8: 1, 10: 1}```
>
> The distribution above reveals that:
> - **93 datasets** contain no mention of failure modes.
> - Only **42 datasets** describe more than one failure mode.
> - Just **6 datasets** include four or more failure modes.
> This underscores a critical limitation in available benchmarks: **limited support for multi-failure-mode scenarios**, which are common in operational industrial systems.
>
> -----
>
> **Comment: 2** What are the reasons behind choosing 10 industrial assets?
>
> **Response**:
>
> In this work we cover 10 assets that are core for the operation of many critical businesses.
> For instance:
> - Power Transformers can be found in Data Centers, National Grid for energy distribution, Heavy Industry, and Renewable Energy plants
> - Aero gas turbines are found in aviation industry, Power Generation, Oil and Gas
> - Electric generators can be found in Hospitals, Data Centers, Telecom, and Construction Sites
>
> We have SMEs which helped in the development and correctness verification of this dataset on these assets with decades of experience on these asset classes. The procedure can be generalizable into any asset classes, as long as we get access to SMEs of that particular asset class.
>
> We started with 51 asset classes out of 800. Our business unit has a library called **Reliability Strategy Library (RSL)**, which provides access to 800 asset classes of varying characteristics. We then narrowed down the selection to 10 asset classes based on field expertise and consultation with subject matter experts (SMEs). Our dataset covers in total 56 unique failure modes and 28 unique sensors, with average of 11.8 sensors per asset and 11.1 failure modes per asset. Additionally, we have work in progress for adding another 35 asset classes
>
> -----
>
> **Comment: 3** What are some asset-specific challenges?
>
> **Response**:
>
> There are several factors that contribute to the complexity of the question:
> - **Asset Complexity**: We can consider the difficulty of the questions in the asset level. Assets have a varying number of connected components and subsystems and their dynamics. For instance, Steam Turbine has many components and requires knowledge in thermodynamics, materials science, and rotor dynamics.
> - **Knowledge Gaps**: Alternatively, is the complexity due to limited understanding of specific failure modes, sensor interactions, or component behaviors?
> - **Amount of reasoning** needed to combine knowledge and infer relationships:  Combining knowledge with logical deductions to answer. Example reasoning in appendix figure 10-12.
>
> These questions point towards a deeper need to formalize how we quantify question complexity in industrial diagnostics.
>
> -----
>
> **Comment: 4** I am curious why the authors claim FailureSensorIQ is a benchmarking "system," not just a dataset.
>
> **Response**:
>
> Our system focuses on multiple aspects of reasoning. We have developed a benchmarking pipeline described in **Section 3** which provides insights on the LLM’s knowledge and reasoning capabilities for our task from different lens using Perturbation-Uncertainty-Complexity analysis (PUC). The benchmarking system is very easy to run. This pipeline supports **repeatable evaluation and benchmarking**—not only for our dataset but also for external ones.
>
> **Task 1: Model Revision Automation**
>
> Given a new model or model release, our infrastructure enables seamless evaluation. A demonstration of this on the new Qwen3-8B model: ```python run_eval.py Qwen/Qwen3-8B full```
>
> - **Main results (Similar to table E.1)**
>
> | Model | Acc_Overall  | Acc_Sel. | Acc_El. | Acc_Perturb | Acc_Consist |
> |--|-|-|-|-|-|
> | Qwen3-8B | 52.01 | 49.57 | 58.60 | 51.48 | 41.62 |
>
> - **Asset performance analysis (Similar to table 11)**
>
> | Asset | Accuracy |
> |-|-|
> | steam turbine | 65.38 |
> | aero gas turbine | 43.86 |
> | industrial gas turbine | 42.56 |
> | pump | 62.92 |
> | compressor | 50.00 |
> | reciprocating IC engine | 58.33 |
> | electric generator |62.82 |
> | fan | 53.50 |
> | power transformer | 44.85 |
>
> - **Uncertainty quantification (Similar to Table 3)**
>
> | UAcc | Set Size | Coverage Rate | Accuracy |
> |-|-|-|-|
> | 17.88 | 3.74 | 92.13 |27.26 |
>
>
> The pipeline outputs a results file which is used by the Hugging Face leaderboard. This simplifies the evaluation process, turning model revision into a reproducible task.
>
> **Task 2: Model and Prompting Strategy Selection**
>
> For example:
>
> - When should one use **Chain-of-Thought prompting on a smaller model** vs. **direct prompting on a larger model**?
> - How does repeated usage impact performance or hallucination?
> - What heuristics can guide this selection?
>
>
>
> **Task 3: PUC Analysis (Perturbation–Uncertainty–Complexity)**
> To assess LLM's knowledge and reasoning capacities from different lenses, propose the **PUC** framework:
> - Perturbation: How does the model respond to slightly altered inputs?
> - Uncertainty: How confident is the model in its response?
> - Complexity: How does task or question complexity correlate with performance?
> To the best of our knowledge, this is the first work proposing **PUC as a unified evaluation paradigm**.
>
> We envision a future where LLMs are treated as **first-class citizens** in industrial AI systems.

---

> > ### Author Response · Authors · 2025-08-06
> >
> > Thank you very much for your thoughtful review and for supporting our submitted work. We truly appreciate your time and insights.
> >
> > We are writing to kindly seek your input on our rebuttal response and to inquire whether any further clarification or action is needed from our side. Based on additional feedback, we remain committed to improving the submission accordingly.

---

> > ### Comment · Reviewer_R1pw · 2025-08-06
> >
> > Thank you for the detailed clarification. Please incorporate our discussion into the camera-ready manuscript. I have raised my score.

---

> > > ### Author Response · Authors · 2025-08-06
> > >
> > > We sincerely appreciate your support for this specialized domain, which deserves greater attention. We will be sure to incorporate our discussions and your feedback into the camera-ready manuscript.

---

### Note · Authors · 2025-08-12

We thank all the reviewers for providing solid technical feedback on our work.
The reviewers acknowledge the strengths of the paper:
- Its criticality in addressing a gap of LLM capabilities in Industry 4.0
- The rigorous analysis conducted for assessing the LLM's capabilities from different lens (Perturbation, Uncertainty, Complexity)
- The novel tasks introduced
- Reproducibility and openness of the evaluation framework
- Clear presentation and easy to understand
- Involving domain-expert for evaluation
- ReAct experiment uncovers misalignment between external retrieval and performance gain

We have also addressed all the concerns by the reviewers which got acknowledged by them. We will be incorporating in the camera-ready version.

---

### Decision · Program_Chairs · 2025-09-18

**Decision:**

Accept (poster)

**Comment:**

The work is the first benchmark on sensor–failure mode reasoning for industrial diagnostics. The benchmark is well-constructed, combining ISO-based mappings, expert validation, and a suite of robustness and uncertainty analyses across 10 industrial assets. The experiments are strong with insights into LLM fragility under perturbations and their limitations in safety-critical domains. While the dataset is semi-synthetic and scope-limited, the contribution is valuable for future research.